# DIFFUSION POLICY THROUGH CONDITIONAL PROXIMAL POLICY OPTIMIZATION

## ABSTRACT

Reinforcement learning (RL) has been extensively employed in a wide range of decision-making problems, such as games and robotics. Recently, diffusion policies have shown strong potential in modeling multi-modal behaviors, enabling more diverse and flexible action generation compared to the conventional Gaussian policy. Despite various attempts to combine RL with diffusion, a key challenge is the difficulty of computing action log-likelihood under the diffusion model. This greatly hinders the direct application of diffusion policies in on-policy reinforcement learning. Most existing methods calculate or approximate the log-likelihood through the entire denoising process in the diffusion model, which can be memory- and computationally inefficient. To overcome this challenge, we propose a novel and efficient method to train a diffusion policy in an on-policy setting that requires only evaluating a simple Gaussian probability. This is achieved by aligning the policy iteration with the diffusion process, which is a distinct paradigm compared to previous work. Moreover, our formulation can naturally handle entropy regularization, which is often difficult to incorporate into diffusion policies. Experiments demonstrate that the proposed method produces multimodal policy behaviors and achieves superior performance on a variety of benchmark tasks in both IsaacLab and MuJoCo Playground.

## 1 INTRODUCTION

Recently, diffusion models have been introduced into reinforcement learning (RL) as diffusion policies (Chi et al., 2023) and have achieved great success due to their strong generative capabilities. Unlike conventional Gaussian policies, computing gradients of the reward with respect to diffusion network parameters is often computationally expensive or even intractable when using standard policy gradient methods, posing significant challenges for policy optimization. In this work, we focus on the on-policy setting and propose a novel framework that trains diffusion policies efficiently.

Early work, such as Wang et al. (2022; 2024), calculates the diffusion policy's gradient using the reparameterization trick, i.e., applying the chain rule through the Q-value and the entire denoising process. However, this approach is often invalid for on-policy algorithms, where a non-differentiable advantage function estimation is utilized rather than a differentiable Q-value estimation. To enable gradient computation via log-likelihood in the on-policy setting, Ding et al. (2025) leverages exact diffusion inversion to reformulate the diffusion process as an invertible deterministic mapping. The log-likelihood can be directly calculated by the change of variables, like in *normalizing flow*. However, as they mentioned, such methods are very computationally expensive due to the recursive nature of the multi-step-based transformation. This is also a common issue in normalizing-flow-style transformation method (Ding et al., 2025; Chao et al., 2024; Chen et al., 2018): authors typically keep the number of steps (or the model depth) relatively small to balance accuracy and computational efficiency. Recently, McAllister et al. (2025) proposed to utilize the flow matching loss to approximate the log-likelihood ratio; however, this method can not handle the entropy regularization term, which is important for exploration in RL (Haarnoja et al., 2018).

Different from the previous work, we overcome the difficulty of computing the diffusion policy's log-likelihood by a different diffusion policy parameterization method. We argue that the policy iteration and the diffusion generative process can be well-aligned. Every policy iteration can be regarded as a denoising step in the diffusion model, and the whole policy iteration process forms

- Diffusion Generative Process

$$p_{\text{prior}} = p^0 \to \cdots \to p^k \to p^{k+1} \to \cdots \to p^N \approx p_{\text{data}}$$

$$\Downarrow$$

$$p^{k+1}(x) = \int p^k(x_0) \underbrace{\mathcal{N}(x|x_0; \mu_d(x_0, k), \Sigma_d(x_0, k))}_{\text{Gaussian}} \, dx_0$$

$$\downarrow \text{ determined by}$$

Reverse SDE
$$x_{k+1} = x_k + \Delta t \cdot \bar{f}(x_k, k) + g(x_k, k)\sqrt{\Delta t} \cdot \mathcal{N}(0, I)$$
or Langevin Dynamics
$$x_{k+1} = x_k + \Delta t' \cdot \frac{1}{2}\nabla_{x_k} \log p_k(x_k) + \sqrt{\Delta t'} \cdot \mathcal{N}(0, I)$$

- Proposed Policy Iteration in RL

$$\pi_{\text{init}} = \pi^0 \to \cdots \to \pi^k \to \pi^{k+1} \to \cdots \to \pi^N \approx \pi_{\text{optimal}}$$

$$\Downarrow$$

$$\pi^{k+1}(a|s) = \int \pi^k(a_0|s) \underbrace{p_\theta(a|a_0, s)}_{\text{Gaussian}} \, da_0$$

$$\downarrow \text{ determined by}$$

**Conditional PPO**

$$\mathbb{E}_{s \sim p_{\pi_\theta}, a_0 \sim \tilde{\pi}, a \sim p_\theta(a|a_0, s)} \left[ \hat{A}^{\pi_{\theta_{\text{sample}}}}(s, a) \right]$$

Figure 1: Overview of the proposed method. We align reinforcement learning policy iteration with the diffusion generative process through a novel policy parameterization. Unlike standard diffusion, where the probability density function is updated via a pre-defined SDE Euler–Maruyama step, our method employs conditional PPO to determine the Gaussian kernel for policy updates.

the whole denoising process. To this end, we propose a novel framework to train a diffusion policy in an on-policy setting; every policy iteration only requires solving a Gaussian policy improvement problem, followed by a diffusion policy fitting with known samples, see Fig. 1 for an overview. The proposed method avoids directly computing the log-likelihood of the diffusion model; instead, it only requires the log-likelihood of a Gaussian distribution, making the implementation highly efficient.

The contribution of this work can be summarized as follows:

1. We propose an on-policy reinforcement learning framework with a novel diffusion policy parameterization method, which closely couples the policy iteration with the diffusion generative process, providing a new perspective to train a diffusion policy.

2. Every policy iteration is converted into a conventional Gaussian policy improvement problem, which can be solved very efficiently, avoiding the expensive computation of a log-likelihood for a diffusion model. Based on such a conversion, the proposed framework can naturally handle the entropy regularization.

3. We evaluate our method in multiple simulation scenarios, illustrating the multi-modality expressive capability and superior performance in multiple benchmark problems.

The code will be open-sourced after the final decision.

## 2 BACKGROUND

### 2.1 REINFORCEMENT LEARNING

Reinforcement learning is formalized as a Markov Decision Process (MDP). The objective is to learn a policy $\pi(\boldsymbol{a}|\boldsymbol{s})$ that determines actions conditioned on states to maximize the expected discounted return $\mathbb{E}_{\tau \sim \pi}[R(\tau)]$, where $\tau := (\boldsymbol{s}_k, \boldsymbol{a}_k)_{k \geq 0}$ denotes a trajectory generated by the agent interacting with the environment under policy $\pi$. This problem can be solved by policy improvement, that is, find a new policy $\pi(\boldsymbol{a}|\boldsymbol{s})$ compared to a reference policy $\tilde{\pi}(\boldsymbol{a}|\boldsymbol{s})$ such that the objectives of the RL problem do not decrease. The new policy can be obtained by solving the following optimization:

$$\max_\pi \mathbb{E}_{\boldsymbol{s} \sim p_\pi(\boldsymbol{s}), \boldsymbol{a} \sim \pi(\boldsymbol{a}|\boldsymbol{s})} \left[ A^{\tilde{\pi}}(\boldsymbol{s}, \boldsymbol{a}) \right] \tag{1}$$

where $p_\pi(\boldsymbol{s})$ is the discounted state visitation distribution under policy $\pi$ (Schulman et al., 2015a; Frans et al., 2025a). $A^{\tilde{\pi}}(\boldsymbol{s}, \boldsymbol{a}) = Q^{\tilde{\pi}}(\boldsymbol{s}, \boldsymbol{a}) - V^{\tilde{\pi}}(\boldsymbol{s})$ is the advantage function, which is the Q function minus the value function under the reference policy. In practice, we usually utilize $p_{\tilde{\pi}(\boldsymbol{s})}$ instead of $p_\pi(\boldsymbol{s})$ due to the difficulty of obtaining unknown $p_\pi(\boldsymbol{s})$. This approximation requires that

the new policy be close to the reference policy, resulting in the following optimization, which is the Proximal Policy Optimization (PPO) (Schulman et al., 2017):

$$\max_{\pi} \mathbb{E}_{\boldsymbol{s} \sim p_{\tilde{\pi}(\boldsymbol{s})}, \boldsymbol{a} \sim \tilde{\pi}(\boldsymbol{a}|\boldsymbol{s})} \left[ \min \left( \frac{\pi(\boldsymbol{a}|\boldsymbol{s})}{\tilde{\pi}(\boldsymbol{a}|\boldsymbol{s})} \hat{A}^{\tilde{\pi}}(\boldsymbol{s}, \boldsymbol{a}), \mathrm{clip}(\frac{\pi(\boldsymbol{a}|\boldsymbol{s})}{\tilde{\pi}(\boldsymbol{a}|\boldsymbol{s})}, 1 - \varepsilon, 1 + \varepsilon) \hat{A}^{\tilde{\pi}}(\boldsymbol{s}, \boldsymbol{a}) \right) \right]. \quad (2)$$

The clip trick aims to keep the new policy close to the reference policy in policy improvement. $\hat{A}^{\tilde{\pi}}$ is the estimated advantage function using methods like generalized advantage estimation (GAE) (Schulman et al., 2015b). Repeating solving optimization equation 2 until convergence leads to an on-policy learning procedure. However, this formulation typically requires the policy to have an analytical form, since solving the optimization involves evaluating the gradient of $\pi_\theta(\boldsymbol{a}|\boldsymbol{s})$. This results in a challenge for diffusion-based policies, as evaluating the probability or its gradient of a diffusion model is often computationally expensive or even intractable.

## 2.2 DIFFUSION GENERATIVE MODEL

Diffusion models generate samples from the target distribution by a denoising process. This can be described by a stochastic differential equation (SDE) with both the forward and the corresponding reverse processes (Song et al., 2021). A data distribution $p_{\mathrm{data}}(\boldsymbol{x})$ can be diffused via the following SDE to a prior distribution $p_{\mathrm{prior}}(\boldsymbol{x})$:

$$d\boldsymbol{x} = f(t)\boldsymbol{x}dt + g(t)d\boldsymbol{w}. \quad (3)$$

where $d\boldsymbol{w}$ is the Wiener process, $f(t)$ and $g(t)$ are predefined coefficients to let the distribution usually converge or be close to a Gaussian distribution $\boldsymbol{x}_T \sim p_{\mathrm{prior}}(\boldsymbol{x})$. In this process, the score function $\nabla_{\boldsymbol{x}} \log p_t(\boldsymbol{x})$ can be learned using a network by score matching (Song & Ermon, 2019; Song et al., 2021). Then solving the corresponding reversed SDE

$$d\boldsymbol{x} = [f(t)\boldsymbol{x} - g(t)^2 \nabla_x \log p_t(\boldsymbol{x})]dt + g(t)d\bar{\boldsymbol{w}} \quad (4)$$

from $\boldsymbol{x}_T$ to $\boldsymbol{x}_0 \sim p_{\mathrm{data}}(\boldsymbol{x})$ results in target sample generation. When solving the reverse SDE by the Euler–Maruyama method, the generative process can be expressed as

$$p_{t_0}(\boldsymbol{x}) = \int p(\boldsymbol{x}_{t_0}|\boldsymbol{x}_{t_1}) \int \cdots \int p(\boldsymbol{x}_{t_{T-1}}|\boldsymbol{x}_{t_T}) p(\boldsymbol{x}_{t_T}) d\boldsymbol{x}_{t_T} \cdots d\boldsymbol{x}_{t_1} \quad (5)$$

where $t_k$ represents the discrete time step, $p(\boldsymbol{x}_{t_{T-1}}|\boldsymbol{x}_{t_T})$ is a Gaussian distribution determined by the score function and predefined coefficients. This can be regarded as the case of DDPM (Ho et al., 2020). In the generative process, since the score function is known, we can also utilize the Langevin dynamics to correct the solution of a numerical SDE solver (Song et al., 2021). This can be expressed by

$$\boldsymbol{x}_{t_k} \leftarrow \boldsymbol{x}_{t_k} + \Delta t \cdot \frac{1}{2} \nabla_{\boldsymbol{x}} \log p_{t_k}(\boldsymbol{x}_{t_k}) + \sqrt{\Delta t} \cdot \boldsymbol{z} \qquad \boldsymbol{z} \sim \mathcal{N}(\boldsymbol{0}, \boldsymbol{I}). \quad (6)$$

With a sufficiently small $\Delta t$ and enough steps, the distribution will gradually converge to $p_{t_k}$. This correction process can be applied at any time step $t_k$ in the reverse SDE to adjust the distribution $p_{t_k}$, compensating for the numerical error in solving the SDE. Therefore, the diffusion process can be a combination of the reversed SDE and the Langevin correct steps. In both processes of equation 5 and equation 6, we can observe that the distribution is updated via a conditional Gaussian kernel determined by the score function, which can be unified in Fig. 1. This insight provides a foundation for constructing diffusion policy in the reinforcement learn, see Sec. 3.1.

## 3 CONDITIONAL PROXIMAL POLICY OPTIMIZATION

This section introduces our main contribution. We first propose a novel diffusion policy parameterization that aligns policy iteration with the diffusion process, and then leverage this parameterization to realize policy improvement through a conditional PPO framework.

### 3.1 DIFFUSION POLICY PARAMETRIZATION

Inspired by the diffusion model, we argue that the multiple policy improvement processes can be aligned with the diffusion generative process. To see this, we parameterize the new policy based on the reference policy:

$$\pi_\theta(\boldsymbol{a}|\boldsymbol{s}) = \int \tilde{\pi}(\boldsymbol{a}_0|\boldsymbol{s})p_\theta(\boldsymbol{a}|\boldsymbol{a}_0, \boldsymbol{s})d\boldsymbol{a}_0. \tag{7}$$

The reference policy $\tilde{\pi}(\boldsymbol{a}_0|\boldsymbol{s})$ can be any distribution, not restricted to a Gaussian distribution, as long as it can be sampled, such as a diffusion model. $p_\theta(\boldsymbol{a}|\boldsymbol{a}_0, \boldsymbol{s})$ is modeled as a Gaussian distribution that has the form of

$$p_\theta(\boldsymbol{a}|\boldsymbol{a}_0, \boldsymbol{s}) = \mathcal{N}(\boldsymbol{a}; \boldsymbol{a}_0 + \boldsymbol{\mu}_\theta(\boldsymbol{a}_0, \boldsymbol{s}), \boldsymbol{\Sigma}_\theta(\boldsymbol{a}_0, \boldsymbol{s})). \tag{8}$$

The residual formulation mimics the procedure of numerically solving the SDE. In this sense, the mean and covariance of the Gaussian kernel correspond to the score function-related term and the Wiener process term, respectively, in the reverse SDE or Langevin dynamics. The idea is shown in Fig. 1, we try to model the policy iteration process as a diffusion generative process, with the policy update being realized by a Gaussian kernel convolution. The difference is that in policy iteration, the Gaussian kernel is learned by a conditional PPO (see Sec. 3.2), which only tells how to improve the policy, while in the diffusion model, the Gaussian kernel is predefined to transfer the distribution to a predesignated one.

### 3.2 POLICY IMPROVEMENT

With the policy parameterization in equation 7, we consider the following optimization to achieve policy improvement (Schulman et al., 2015a) in RL:

$$\max_\theta \mathbb{E}_{\boldsymbol{s} \sim p_{\pi_\theta}, \boldsymbol{a} \sim \pi_\theta(\boldsymbol{a}|\boldsymbol{s})} \left[ \hat{A}^{\pi_{\theta_{\text{sample}}}}(\boldsymbol{s}, \boldsymbol{a}) \right] \tag{9}$$

where $\pi_{\theta_{\text{sample}}}$ denotes the policy used for data collection. However, directly optimizing this objective using importance sampling like equation 2 poses a challenge: computing the gradient of the loss is difficult due to the intractability of evaluating $\nabla_\theta \pi_\theta(\boldsymbol{a}|\boldsymbol{s})$ under the parameterization in equation 7. To address this issue, we propose solving a simpler optimization as follows to get the same optimal solution as equation 9:

$$\max_\theta \mathbb{E}_{\boldsymbol{s} \sim p_{\pi_\theta}, \boldsymbol{a}_0 \sim \tilde{\pi}, \boldsymbol{a} \sim p_\theta(\boldsymbol{a}|\boldsymbol{a}_0, \boldsymbol{s})} \left[ \hat{A}^{\pi_{\theta_{\text{sample}}}}(\boldsymbol{s}, \boldsymbol{a}) \right]. \tag{10}$$

In this formulation, actions $\boldsymbol{a} \sim \pi_\theta(\boldsymbol{a}|\boldsymbol{s})$ are sampled by first sampling via $\boldsymbol{a}_0 \sim \tilde{\pi}$, followed by sampling $\boldsymbol{a}$ from the conditional distribution $p_\theta(\boldsymbol{a}|\boldsymbol{a}_0, \boldsymbol{s})$. An immediate advantage of this formulation is that the gradient of the objective becomes easy to compute, due to the Gaussian parameterization of $p_\theta(\boldsymbol{a}|\boldsymbol{a}_0, \boldsymbol{s})$. Notably, the objective function defined in equation 10 is identical to that in equation 9 (see Appendix B for proof). This equivalence allows us to obtain the improved policy through a simple optimization instead of the original intractable formulation. Based on the objective function in equation 10, we can then construct the corresponding surrogate loss as follows:

$$\max_\theta \mathbb{E}_{\boldsymbol{s} \sim p_{\pi_{\theta_{\text{sample}}}}, \boldsymbol{a}_0 \sim \tilde{\pi}, \boldsymbol{a} \sim p_{\theta_{\text{sample}}}(\boldsymbol{a}|\boldsymbol{a}_0, \boldsymbol{s})} \left[ \text{CLIP} \left( \frac{p_\theta(\boldsymbol{a}|\boldsymbol{a}_0, \boldsymbol{s})}{p_{\theta_{\text{sample}}}(\boldsymbol{a}|\boldsymbol{a}_0, \boldsymbol{s})} \hat{A}^{\pi_{\theta_{\text{sample}}}}(\boldsymbol{s}, \boldsymbol{a}) \right) \right], \tag{11}$$

where CLIP is a shorthand for clipped loss defined in equation 2. The clip trick here aims to constrain $p_\theta(\boldsymbol{a}|\boldsymbol{a}_0, \boldsymbol{s})$ not to be far away from $p_{\theta_{\text{sample}}}(\boldsymbol{a}|\boldsymbol{a}_0, \boldsymbol{s})$. In our approach, actions are sampled from a conditional Gaussian distribution, effectively transforming the original policy optimization problem into the standard PPO framework. We hereby name this formulation as *Conditional Proximal Policy Optimization* (CPPO).

Once we learn the $p_{\theta^*}(\boldsymbol{a}|\boldsymbol{a}_0, \boldsymbol{s})$, we can collect every Gaussian kernel in each policy improvement to form the final diffusion policy. However, this approach becomes increasingly inefficient since the number of diffusion steps grows with the number of policy iterations. To address this issue, we instead utilize a single diffusion model to fit $\pi_{\theta^*}(\boldsymbol{a}|\boldsymbol{s})$ in equation 7 after every policy improvement. Although fitting the diffusion model introduces numerical errors (i.e., $\pi_{\text{diffusion}} \approx \pi_{\theta^*}$), the subsequent policy iteration always samples based on $\pi_{\text{diffusion}}$. As a result, each policy improvement is based on the fitted diffusion model itself. This prevents the fitting error from accumulating to

next iteration. In our experiments, we also observe that the proposed method is robust to this fitting error, see Appendix D.3. For simplicity, we use a flow matching (Lipman et al., 2023) instead of a diffusion model to train the policy in this work. The condition $s$ is encoded in a classifier-free style, i.e., we directly embed the state into a network to approximate the vector field $u_\eta(a, t, s)$, then use the following optimization to train the flow matching:

$$\min_\eta \mathbb{E}_{s,a_1 \sim \mathcal{D}, t \sim \mathcal{U}(0,1)}[\|u_\eta(a_t, t, s) - (a_1 - a_n)\|_2^2] \quad \text{with} \quad a_t = (1-t)a_n + ta_1 \quad (12)$$

where $a_n$ is the sample from $\mathcal{N}(0, I)$, $(s, a_1)$ are sampled from $\pi_{\theta^*}$. The generating process can be achieved by solving the ODE $dx = u_\eta(a_t, t, s)dt$ from $t = 0$ to $t = 1$, where $a_0 \sim \mathcal{N}(0, I)$.

Repeating policy parameterization and solving the policy improvement constructs the whole policy iteration process. However, the proposed CPPO only improves the policy from $\pi_{\theta_{\text{sample}}}$ to $\pi_{\theta^*}$, rather than from the last optimal $\tilde{\pi}$ to the current optimal $\pi_{\theta^*}$. This breaks the monotonically increasing properties in the entire policy iteration, as we cannot ensure whether $\pi_{\theta_{\text{sample}}}$ is better than $\tilde{\pi}$. To address this issue, we utilize the exponential moving average (EMA) technique on diffusion policy and initialize $p_{\theta_{\text{sample}}}$ as the last optimal $p_{\theta^{k-1}}$ to ensure $\pi_{\theta_{\text{sample}}} \approx \tilde{\pi}$. The parameters of the diffusion policy are updated by $\theta^- \leftarrow (1-\alpha)\theta + \alpha\theta^-$, where $\theta^-$ is the EMA model parameter and $\theta$ is the current parameters of the diffusion model. The EMA technique ensures the model parameter will not update far away from the previous one, that is, $\pi^k \approx \pi^{k-1}$. Therefore, after we obtain a diffusion policy $\pi^k = \int \pi^{k-1}(a_0|s)p_{\theta^*_{k-1}}(a|a_0, s)da_0$, the initial sampling policy for next iteration is

$$\pi_{\theta_{\text{sample}}} = \int \pi^k(a_0|s)p_{\theta^*_{k-1}}(a|a_0, s)da_0 \approx \int \pi^{k-1}(a_0|s)p_{\theta^*_{k-1}}(a|a_0, s)da_0 = \pi^k(a|s). \quad (13)$$

The final equality in equation 13 is formally an approximation due to the EMA update; it is presented here solely to aid understanding. By utilizing the EMA technique, the proposed policy iteration can be approximately regarded as monotonically increasing.

### 3.3 REGULARIZATIONS

**Entropy regularization.** Adding an entropy regularization (Haarnoja et al., 2018) in the loss is essential in reinforcement learning, as it encourages exploration and helps avoid getting trapped in local minima. In our formulation, we need to maximize the entropy $\mathcal{H}(\pi_\theta)$. However, unlike a Gaussian policy whose entropy can be analytically calculated, it is usually difficult to directly calculate the entropy of a diffusion policy. Existing methods that approximate (Wang et al., 2024; Dong et al., 2025) or directly calculate (Chao et al., 2024; Ding et al., 2025) the entropy are computationally complex in general. Fortunately, in our proposed framework, we can effectively maximize a lower bound of $\mathcal{H}(\pi_\theta)$, rather than the intractable entropy itself, by

$$\mathcal{H}(\pi_\theta(a|s)) = \mathcal{I}(p(a, a_0|s)) + \mathcal{H}(p_\theta(a|a_0, s)) \quad \Rightarrow \quad \mathcal{H}(\pi_\theta) \geq \mathcal{H}(p_\theta), \quad (14)$$

where we leverage the the *mutual information* identity $\mathcal{I}(q_{x,y}) = \mathcal{H}(q_x) - \mathcal{H}(q_{x|y})$, with the property that $\mathcal{I}(q_{x,y}) \geq 0$ (Cover, 1999). Consequently, instead of directly maximizing the intractable $\mathcal{H}(\pi_\theta)$, we maximize its computable lower bound $\mathcal{H}(p_\theta)$, which only requires evaluating the entropy of a Gaussian distribution. This is more efficient compared to the previous methods for handling the entropy of a diffusion policy.

**Score-Based Regularization.** Although diffusion policy has strong expression ability, its training can be slow and unstable due to the diffusion policy distribution lying in an infinite-dimensional functional space. To address this issue, we introduce an empirical regularization term to accelerate the convergence of the diffusion policy, which is given by

$$\min_\theta \mathcal{R}(\theta) = \mathbb{E}_{s \sim p_{\pi_{\theta_{\text{sample}}}}, a_0 \sim \tilde{\pi}}[\|\mu_\theta(a_0, s) - \frac{1}{2}\Sigma_\theta(a_0, s)(-a_0)\|_2^2] \quad (15)$$

where $\mu_\theta$ and $\Sigma_\theta$ are defined in equation 8, $-a_0$ is the score function $\nabla_x \log \mathcal{N}(0, I)$. This regularization term aims to let the diffusion policy converge to a standard Gaussian distribution. The basic idea is to let $\mu_\theta$ align with the score function of a standard Gaussian such that the sample path $a = a_0 + \delta a_0$ in equation 8 be a Langevin dynamics update toward the standard Gaussian, see Appendix C for details. This regularization imposes a KL-like constraint that prevents the policy from drifting too far from the prior distribution. We illustrate that this regularization accelerates and

stabilizes the training process in Sec. 4.4. Although this score-based regularization introduces an inductive bias to the diffusion policy, the multi-modality expression ability will not degenerate since the major objective function is reinforcement learning objectives, which can be illustrated in Fig. 2, where the task is trained by incorporating this regularization.

By arranging all regularization terms into the RL objectives, we can obtain the final optimization as

$$\max_{\theta} \mathbb{E}_{\boldsymbol{s}\sim p_{\pi_\theta},\boldsymbol{a}_0\sim\tilde{\pi},\boldsymbol{a}\sim p_\theta(\boldsymbol{a}|\boldsymbol{a}_0,\boldsymbol{s})} \left[ \hat{A}^{\pi_{\theta_\text{sample}}}(\boldsymbol{s},\boldsymbol{a}) + \alpha\mathcal{H}(p_\theta) - \beta\mathcal{R}(\theta) \right]. \tag{16}$$

where $\alpha$ and $\beta$ are the coefficients of the regularizations. The proposed method can be summarized in Algorithm 1.

---

**Algorithm 1** DP-CPPO: Diffusion Policy through Conditional PPO

---

**Require:** Initial flow policy parameters $\eta$, value function parameters $\phi$, residual policy parameters $\theta$, other hyperparameters.
1: **for** $i = 1$ to $N$ **do**
2:     Collect state-action trajectories $\mathcal{D} = \{s, a_0, a\}$ using $\tilde{\pi}_\eta$ and $p_\theta$; estimate $\hat{A}$.
3:     Update the residual policy $p_\theta$ via a PPO-style optimization:

$$\theta \leftarrow \max_{\theta} \mathbb{E}_{(\boldsymbol{s},\boldsymbol{a}_0,\boldsymbol{a})\sim\mathcal{D}} \left[ \text{CLIP}\left( \frac{p_\theta(\boldsymbol{a}|\boldsymbol{a}_0,\boldsymbol{s})}{p_{\theta_\text{sample}}(\boldsymbol{a}|\boldsymbol{a}_0,\boldsymbol{s})} \hat{A}^{\pi_{\theta_\text{sample}}}(\boldsymbol{s},\boldsymbol{a}) \right) + \alpha\mathcal{H}(p_\theta) - \beta\mathcal{R}(\theta) \right]$$

4:     Sample the flow action $\boldsymbol{a}_1$ via $\tilde{\pi}_\eta$ followed by $p_\theta$, update flow policy parameter $\eta$ by:

$$\min_{\eta} \mathbb{E}_{\boldsymbol{s},\boldsymbol{a}_1\sim\mathcal{D},t\sim\mathcal{U}(0,1)}[\|\boldsymbol{u}_\eta(\boldsymbol{a}_t,t,\boldsymbol{s}) - (\boldsymbol{a}_1 - \boldsymbol{a}_n)\|_2^2]$$

5:     Update value function parameters $\phi$ like a standard PPO.
6: **end for**
7: **return** Flow policy $\tilde{\pi}_\eta$.

---

## 4 EXPERIMENTS

In this section, we aim to illustrate that 1) the proposed method is able to learn a diffusion policy with multi-modal expressiveness; 2) the proposed method can be trained efficiently, and the entropy regularization benefits the performance; 3) the proposed method shows superior results on several IsaacLab (Mittal et al., 2023) and Playground (Zakka et al., 2025) benchmarks. For comparison, we restrict our comparison to other *on-policy* algorithms: (a) the standard Gaussian PPO, and (b) a diffusion policy named FPO (McAllister et al., 2025). Our method is denoted as DP-CPPO.

### 4.1 MULTI-MODALITY ILLUSTRATION

An environment named "Multi-Goal" (Haarnoja et al., 2017; Dong et al., 2025) is used to demonstrate that the proposed diffusion policy achieves multimodal behavior. The objective for the agent is to reach any goal in the map, see Fig. 2. The main reward is defined by the distance cost of the agent's position to the nearest goal. Besides, when the agent reaches any goal, an extra single reward will be assigned. As the goals are set symmetrically, an agent starting at a saddle point should naturally exhibit a multimodal action distribution to represent the possibility of moving toward any goal. The actions at these saddle points, visualized in Fig. 2, demonstrate the multimodal behavior captured by our diffusion policy, overcoming the limitations of unimodal distributions. Moreover, we utilize a Gaussian mixture model analysis that confirms that the number of modes in these action distributions is consistently greater than one.

**Multi-modality and Reward.** Compared to unimodal policies such as the Gaussian policy, a key advantage of the diffusion policy is that its inherent multimodality can lead to higher rewards at saddle points. The rewards are shown in Tab. 1. At these points, actions directed toward opposite goals provide identical rewards.

Table 1: Saddle point reward, higher is better.

| Method | Initial Position | | |
|---|---|---|---|
| | (0.0,0.0) | (2.5,2.5) | (-2.5,2.5) |
| Gaussian PPO | -41.80 | -16.47 | -6.47 |
| DP-CPPO | **-11.94** | **1.82** | **1.49** |

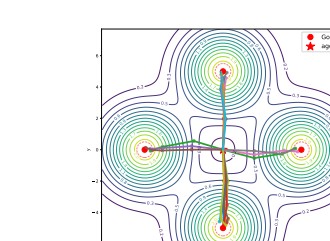
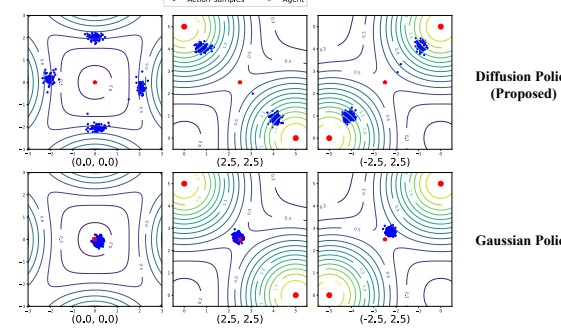

Figure 2: *Left.* Multi-Goal environments, where different trajectories starting from the origin under the diffusion policy are illustrated. Contour lines in the figure are drawn based on the distance cost. *Right.* The positions after taking the first action from different saddle points are shown, visualizing the distribution of the policy $\pi(\boldsymbol{a}|\boldsymbol{s})$. The diffusion policy exhibits multimodal behavior, whereas the Gaussian policy collapses to near-zero movement due to the averaging effect of opposite goals.

A unimodal distribution, however, tends to average the gradients toward each goal, collapsing into a degenerate solution (e.g., no movement; see Fig. 2). In contrast, the diffusion policy maintains multimodal behavior, preventing collapse and enabling diverse trajectories that align with the different goals, thereby directly increasing rewards. However, this advantage becomes less pronounced when the true optimal policy is unimodal.

### 4.2 COMPUTATIONAL EFFICIENCY AND ENTROPY REGULARIZATION

**Computational Efficiency.** Unlike GenPo (Ding et al., 2025), which computes the diffusion log-likelihood by backpropagating through the entire denoising process, our framework reduces each update to a conditional Gaussian PPO step followed by flow matching. This formulation provides a lightweight and efficient training pipeline for diffusion policies. As shown in Tab. 2, training for 1K epochs in the IsaacLab Ant task only requires a computational cost comparable to standard PPO. The crucial difference is that our diffusion model is optimized via flow matching, which is simulation-free (Lipman et al., 2023), rather than through a recursive normalizing-flow-style inversion. Memory consumption remains unchanged as the number of flow steps increases, further illustrating this point.

Table 2: Computational cost for Ant task in IsaacLab.

|  | PPO | DP-CPPO (flow step = 8) | DP-CPPO (flow step = 16) |
|---|---|---|---|
| Training time (min) | 4.68 | 8.05 (71.9% ↑) | 9.31 (99.9% ↑) |
| GPU memory (MB) | 4202 | 4306 (2.5% ↑) | 4306 (2.5% ↑) |

**Entropy Regularization.** The entropy term encourages exploration during RL training, helping the policy avoid getting trapped in local minima and enabling higher rewards. Unlike diffusion policies such as FPO (McAllister et al., 2025), which cannot directly incorporate entropy regularization, our method naturally supports it, allowing exploration to be adjusted for improved performance. To demonstrate this, we report results on the Playground FingerSpin task. As shown in Fig. 3, removing the entropy term yields results similar to FPO. However, with a properly chosen scale, the reward increases significantly. In contrast, an excessively large scale (e.g., 0.05) will destabilize the training, resulting in near-zero rewards as the policy diverges in this case.

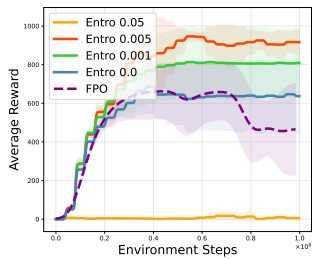

Figure 3: Rewards on Playground FingerSpin.

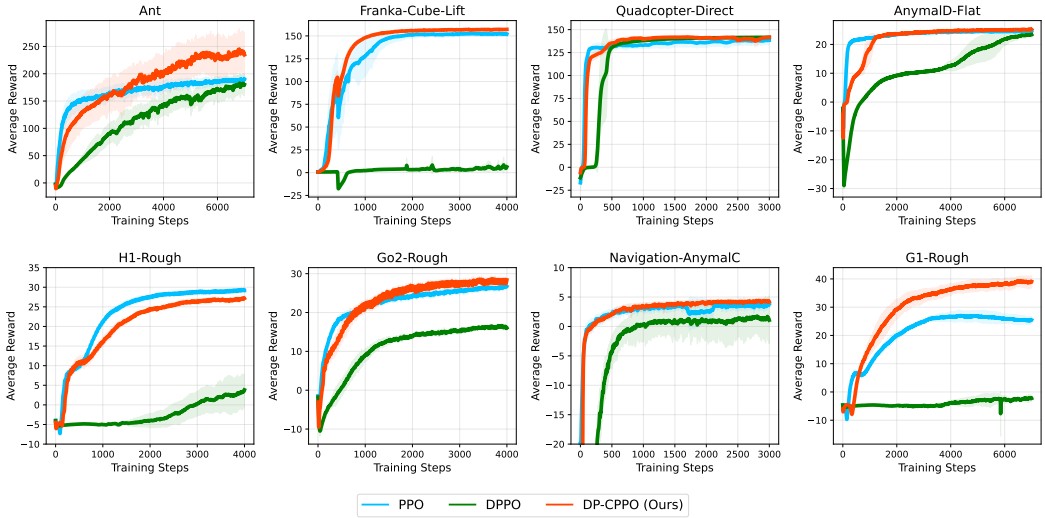

Figure 4: Training rewards across eight environments in IsaacLab. Results show the mean and standard deviation over 5 runs with different seeds (higher is better). For better visualization, we have smoothed the reward data. The proposed method outperforms Gaussian PPO in most tasks.

## 4.3 RESULTS ON BENCHMARKS

We also evaluate the proposed method in several other benchmark control tasks in IsaacLab and Mujoco Playground. Since FPO is implemented in *JAX*, making it inconvenient to deploy directly in Torch-based IsaacLab environments, we compare with PPO in IsaacLab and FPO in Playground. In addition, we include the results of a diffusion policy method DPPO (Ren et al., 2024) in both platforms, which are only provided as a reference since it was originally designed to be fine-tuned on pretrained diffusion policy rather than learning from scratch.

**IssacLab.** We evaluate algorithms on eight benchmarks in IsaacLab. For the Gaussian PPO baseline, we adopt the default RSL-RL implementation (Rudin et al., 2022), using its default hyperparameter settings in IsaacLab. Since our method is built upon PPO, we implement it within the RSL-RL framework, keeping most hyperparameters consistent with the defaults and trying to minimize additional tuning. The final evaluating reward is reported in Tab. 3 and the training curve is in Fig. 4. To ensure statistical reliability, the results are the average of 5 trainings with different random seeds, with the standard deviation reported. In the training process, we utilize a diffusion policy $\pi_{\text{flow}}$ to fit a combined optimal policy $\pi_{\theta*}(a|s) = \int \tilde{\pi}(a_0|s)p_{\theta*}(a|a_0, s)da_0$ in each iteration. To verify that this fitting ultimately converges, we report results for both the diffusion-only policy $\pi_{\text{flow}}$ (denotes "Flow") and the optimal policy $\pi_{\theta*}(a|s)$ (denotes "Flow+Res") in Tab. 3. The results show that diffusion-only and combined policies are very close, illustrating that our method enables the diffusion policy to converge. Moreover, the reward results show that our method achieves slightly higher or comparable final rewards to RSL-RL PPO across most tasks, demonstrating the effectiveness of the proposed method. Additional results and details can be found in Appendix D.1 and D.3.

Table 3: Evaluating rewards on IsaacLab tasks, higher is better.

|  | Ant | Franka | Quadcopter | AnymalD | H1 | Go2 | Navigation | G1 |
|---|---|---|---|---|---|---|---|---|
| Flow | $220.74_{\pm 33.31}$ | $159.43_{\pm 0.37}$ | $141.34_{\pm 0.77}$ | $25.33_{\pm 0.10}$ | $29.40_{\pm 0.41}$ | $34.71_{\pm 0.15}$ | $4.58_{\pm 0.10}$ | $43.89_{\pm 1.32}$ |
| Flow+Res | $\mathbf{248.47}_{\pm 37.03}$ | $\mathbf{159.48}_{\pm 0.42}$ | $\mathbf{141.88}_{\pm 0.36}$ | $\mathbf{25.34}_{\pm 0.14}$ | $29.41_{\pm 0.42}$ | $\mathbf{34.97}_{\pm 0.28}$ | $\mathbf{4.59}_{\pm 0.09}$ | $\mathbf{44.17}_{\pm 1.21}$ |
| PPO | $199.09_{\pm 10.58}$ | $156.46_{\pm 2.28}$ | $138.44_{\pm 4.71}$ | $24.64_{\pm 0.07}$ | $\mathbf{30.63}_{\pm 0.35}$ | $32.56_{\pm 0.20}$ | $4.31_{\pm 0.11}$ | $28.48_{\pm 1.28}$ |
| DPPO | $192.73_{\pm 16.16}$ | $7.76_{\pm 6.34}$ | $141.64_{\pm 0.22}$ | $23.65_{\pm 0.46}$ | $9.07_{\pm 3.70}$ | $16.59_{\pm 0.00}$ | $2.13_{\pm 2.49}$ | $2.87_{\pm 1.43}$ |

**Playground.** We further evaluate the proposed method against another diffusion-based approach, FPO. For a fair comparison, we directly adopt their open-source implementation on Playground as the baseline, while our method is implemented on Playground through a JAX-to-Torch interface. The results are presented in Tab. 4 and Fig. 5. Our method also achieves slightly higher rewards

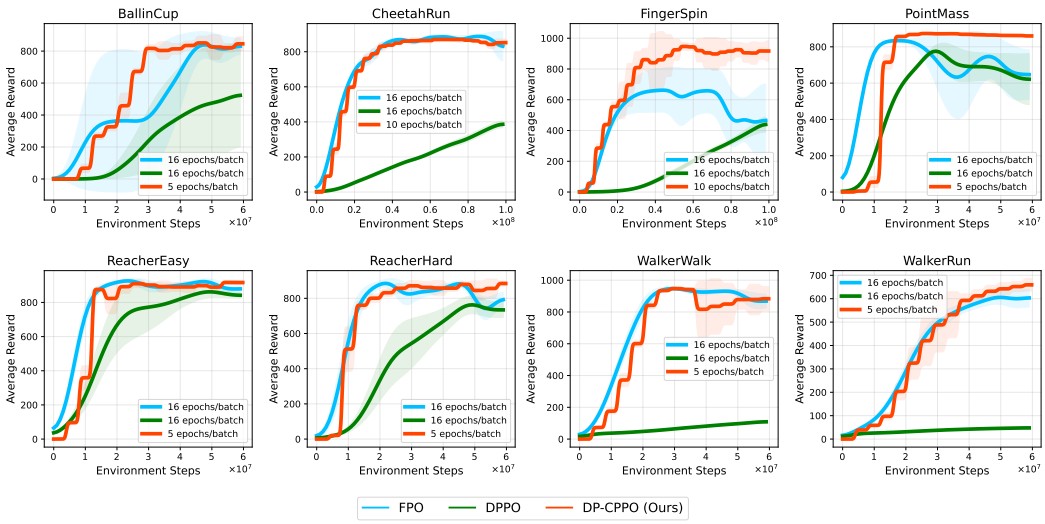

Figure 5: Training rewards across eight environments in Playground. Results show the mean and standard deviation over 5 runs with different seeds (higher is better). For better visualization, we have smoothed the reward data. Here, 'epoch/batch' denotes how many times each rollout is reused; smaller values indicate higher data efficiency. Our method achieves superior data efficiency.

than FPO in most tasks. The difference between the diffusion-only and combined results appears larger than in IsaacLab. This is mainly due to the restricted number of training epochs for a fair comparison with FPO; with longer training, the diffusion policy converges in the same manner as in IsaacLab. Additional results and details can be found in Appendix D.2.

Table 4: Evaluating rewards on Playground tasks, higher is better.

|  | BallCup | Cheetah | FingerSpin | PointMass | Reach-E | Reach-H | WlkWalk | WlkRun |
|---|---|---|---|---|---|---|---|---|
| Flow | $846.8_{\pm 57.7}$ | $822.6_{\pm 26.0}$ | $868.2_{\pm 92.0}$ | $\mathbf{860.3}_{\pm 6.0}$ | $\mathbf{918.9}_{\pm 11.4}$ | $857.2_{\pm 24.7}$ | $865.9_{\pm 89.0}$ | $641.4_{\pm 27.3}$ |
| Flow+Res | $866.2_{\pm 44.6}$ | $848.1_{\pm 19.1}$ | $\mathbf{912.5}_{\pm 49.1}$ | $853.1_{\pm 2.9}$ | $898.4_{\pm 34.5}$ | $\mathbf{873.4}_{\pm 28.7}$ | $\mathbf{894.0}_{\pm 43.5}$ | $\mathbf{660.5}_{\pm 22.3}$ |
| FPO | $\mathbf{879.4}_{\pm 45.7}$ | $\mathbf{890.2}_{\pm 7.8}$ | $655.2_{\pm 147.9}$ | $712.6_{\pm 85.3}$ | $902.5_{\pm 10.4}$ | $870.1_{\pm 35.8}$ | $880.1_{\pm 28.4}$ | $613.2_{\pm 29.2}$ |
| DPPO | $558.3_{\pm 319.7}$ | $401.6_{\pm 17.1}$ | $465.5_{\pm 57.1}$ | $546.3_{\pm 136.4}$ | $828.2_{\pm 29.3}$ | $742.1_{\pm 49.8}$ | $116.2_{\pm 6.4}$ | $49.7_{\pm 4.6}$ |

## 4.4 ABLATION ON SCORE-BASED REGULARIZATION

To demonstrate the effectiveness of the regularization term in equation 15, we conduct an ablation study evaluating its contribution to performance. We present the results of different scale settings in the AnymalB-Flat and Quadcopter tasks of the IsaacLab environment. As shown in Fig. 6a, using a relatively large scale for this term accelerates reward improvement, and we find that a value of 0.1 works well across nearly all tasks. We also observe that removing this term makes training less stable, sometimes resulting in divergence (Fig. 6a) and at other times in collapse (Fig. 6b). Overall, these results highlight the importance of this term for training stability and efficiency.

## 5 RELATED WORK

Generative models, particularly diffusion models and flow matching (Song et al., 2021; Lipman et al., 2023; Karras et al., 2022), have demonstrated remarkable expressiveness across diverse domains. Motivated by these successes, researchers have increasingly explored integrating diffusion models into RL, yielding promising results such as Diffusion Policy (Chi et al., 2023). In the following, we review prior works that combine diffusion models with RL.

**Diffusion policy for online RL.** Online RL algorithms aim to optimize a policy by repeatedly interacting with the environment during training. For this category method, most previous works focus

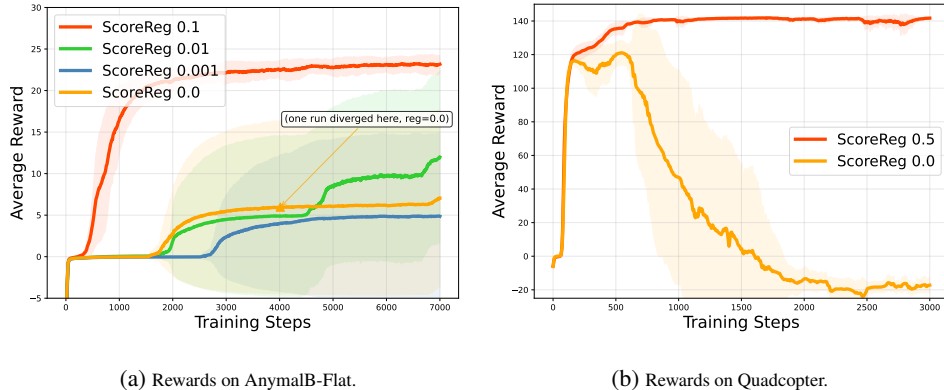

(a) Rewards on AnymalB-Flat.

(b) Rewards on Quadcopter.

Figure 6: Ablation study on score-based regularization.

on off-policy methods, which rely on Q-value estimation. With the Q-value estimated by a network, work like DACER (Wang et al., 2024) directly optimizes the diffusion policy throughout the entire diffusion process via the reparameterization trick, i.e., $\partial \boldsymbol{a}/\partial \theta$. In contrast, works like DIPO (Yang et al., 2023), QVPO (Ding et al., 2024), QSM (Psenka et al., 2023), and MaxEntDP (Dong et al., 2025) incorporate the Q-value information into the diffusion loss to train the diffusion model. These methods heavily rely on the Q-value estimation, whose training process is not always stable. Recently, researchers have tried to train the diffusion policy in an on-policy setting, which requires calculating the log-likelihood under a diffusion model. GenPo (Ding et al., 2025) utilizes the exact diffusion inversion, enabling the log-likelihood computation via change of variables. However, this method is computationally expensive due to its recursive nature, which is similar to propagating the gradient through the whole diffusion process (Wang et al., 2024). FPO (McAllister et al., 2025) approximates the log-likelihood via flow matching loss, which introduces an extra bias. In addition, this method can not handle the entropy regularization. Other work, such as DPPO (Ren et al., 2024), trains diffusion policies by embedding the diffusion MDP directly into the RL MDP. In contrast, our approach seeks to align the two MDPs, representing a fundamentally different formulation.

**Diffusion policy for offline RL.** Offline RL algorithms aim to learn a policy solely from a pre-collected dataset, without any further interaction with the environment. Early work Peng et al. (2019) proposed an analytical solution of an optimal policy for offline RL. Building on this result, sfBC (Chen et al., 2022) selects desirable samples through Q-value weighting and subsequently trains a diffusion policy with the resampled data. More recently, several works have explored leveraging diffusion models with guidance (Dhariwal & Nichol, 2021; Ho & Salimans, 2022) to obtain effective diffusion policies for RL (Lu et al., 2023; Fang et al., 2024; Frans et al., 2025b).

## 6 CONCLUSION

In this work, we propose a novel on-policy algorithm to train the diffusion policy. By aligning the policy iteration with the diffusion process, the diffusion policy is obtained via solving multiple conditional Gaussian PPO problems. This method avoids the log-likelihood computation of a diffusion model, which can be computationally inefficient and even intractable. Within the proposed method, the entropy regularization that encourages exploration also requires only Gaussian entropy calculation. The proposed method provides a simple and efficient way to train a diffusion policy. Experiments show the multimodal behavior of the diffusion policy, which is the key advantage of the diffusion model. We also evaluate the proposed method in multiple benchmarks in IsaacLab and Playground, and the results show that higher rewards can be obtained for the proposed method compared to other on-policy methods.

ETHICS STATEMENT

Our study develops reinforcement learning algorithms in simulation environments. It does not involve human subjects, personal or sensitive data, or applications with foreseeable ethical risks. We have ensured that all experiments comply with the ICLR Code of Ethics. Accordingly, we do not identify any ethical concerns associated with this work.

REPRODUCIBILITY STATEMENT

The implementation code, along with training scripts and configuration files, will be released after the final decision to ensure full reproducibility.

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

## APPENDIX

We present additional derivation, analysis, and experimental results in the Appendix.

## A  USE OF LLMs

This manuscript has benefited from language editing support provided by GPT-5.

## B  CONDITIONAL PPO OBJECTIVE EQUIVALENCE

The key point of this work is to transform a complicated policy optimization problem into a standard PPO formulation, which can be effectively solved (see Sec. 3.2). The equivalence of two objectives can be given by following the idea of *Law of total expectation*:

$$
\begin{aligned}
&\mathbb{E}_{\boldsymbol{s}\sim p_{\pi_\theta},\boldsymbol{a}\sim\pi_\theta(\boldsymbol{a}|\boldsymbol{s})}\left[\hat{A}^{\pi_{\theta_{\text{sample}}}}(\boldsymbol{s},\boldsymbol{a})\right]\\
&=\int p_{\pi_\theta}(\boldsymbol{s})\int\pi_\theta(\boldsymbol{a})\hat{A}^{\pi_{\theta_{\text{sample}}}}(\boldsymbol{s},\boldsymbol{a})d\boldsymbol{a}d\boldsymbol{s}\\
&=\int p_{\pi_\theta}(\boldsymbol{s})\int\left(\int\tilde{\pi}(\boldsymbol{a}_0|\boldsymbol{s})p_\theta(\boldsymbol{a}|\boldsymbol{a}_0,\boldsymbol{s})d\boldsymbol{a}_0\right)\hat{A}^{\pi_{\theta_{\text{sample}}}}(\boldsymbol{s},\boldsymbol{a})d\boldsymbol{a}d\boldsymbol{s}\\
&=\int p_{\pi_\theta}(\boldsymbol{s})\int\int\tilde{\pi}(\boldsymbol{a}_0|\boldsymbol{s})p_\theta(\boldsymbol{a}|\boldsymbol{a}_0,\boldsymbol{s})\hat{A}^{\pi_{\theta_{\text{sample}}}}(\boldsymbol{s},\boldsymbol{a})d\boldsymbol{a}d\boldsymbol{a}_0d\boldsymbol{s}\\
&=\int p_{\pi_\theta}(\boldsymbol{s})\int\tilde{\pi}(\boldsymbol{a}_0|\boldsymbol{s})\int p_\theta(\boldsymbol{a}|\boldsymbol{a}_0,\boldsymbol{s})\hat{A}^{\pi_{\theta_{\text{sample}}}}(\boldsymbol{s},\boldsymbol{a})d\boldsymbol{a}d\boldsymbol{a}_0d\boldsymbol{s}\\
&=\mathbb{E}_{\boldsymbol{s}\sim p_{\pi_\theta},\boldsymbol{a}_0\sim\tilde{\pi},\boldsymbol{a}\sim p_\theta(\boldsymbol{a}|\boldsymbol{a}_0,\boldsymbol{s})}\left[\hat{A}^{\pi_{\theta_{\text{sample}}}}(\boldsymbol{s},\boldsymbol{a})\right].
\end{aligned}
\tag{17}
$$

## C  SCORE-BASED REGULARIZATION

We propose a score-based regularization term in 15 to stabilize and accelerate the diffusion policy training process. Here we explain the motivation and how it works.

In the policy improvement formulation, the policy is given by

$$
\pi_\theta(\boldsymbol{a}|\boldsymbol{s})=\int\tilde{\pi}(\boldsymbol{a}_0|\boldsymbol{s})p_\theta(\boldsymbol{a}|\boldsymbol{a}_0,\boldsymbol{s})d\boldsymbol{a}_0,
\tag{18}
$$

where $p_\theta(\boldsymbol{a}|\boldsymbol{a}_0,\boldsymbol{s})$ is defined by

$$
p_\theta(\boldsymbol{a}|\boldsymbol{a}_0,\boldsymbol{s})=\mathcal{N}(\boldsymbol{a};\boldsymbol{a}_0+\boldsymbol{\mu}_\theta(\boldsymbol{a}_0,\boldsymbol{s}),\boldsymbol{\Sigma}_\theta(\boldsymbol{a}_0,\boldsymbol{s})).
\tag{19}
$$

Therefore, the samples $\boldsymbol{a}^{k+1} \sim \pi_\theta(\boldsymbol{a}|\boldsymbol{s})$ can be obtained by (we add a superscript to indicate the iteration step)

$$\boldsymbol{a}^{k+1} = \boldsymbol{a}^k + \Delta\boldsymbol{a}, \tag{20}$$

where $\boldsymbol{a}^k \sim \tilde{\pi}(\boldsymbol{a}_0|\boldsymbol{s})$ and $\Delta\boldsymbol{a}|\boldsymbol{a}^k \sim \mathcal{N}(\Delta\boldsymbol{a}; \boldsymbol{\mu}_\theta(\boldsymbol{a}^k, \boldsymbol{s}), \boldsymbol{\Sigma}_\theta(\boldsymbol{a}^k, \boldsymbol{s}))$. Note $\Delta\boldsymbol{a}$ is sampled giving $\boldsymbol{a}^k$. Here, we consider the state $\boldsymbol{s}$ as a constant parameter. The process of obtaining samples $\boldsymbol{a}^{k+1}$ can be regarded as an Euler–Maruyama step as follows:

$$\boldsymbol{a}^{k+1} = \boldsymbol{a}^k + \Delta t\frac{\boldsymbol{\mu}_\theta(\boldsymbol{a}^k, \boldsymbol{s})}{\Delta t} + \sqrt{\Delta t}\frac{\boldsymbol{\Sigma}_\theta^{\frac{1}{2}}(\boldsymbol{a}^k, \boldsymbol{s})}{\sqrt{\Delta t}}\boldsymbol{z} \qquad \boldsymbol{z} \sim \mathcal{N}(\boldsymbol{z}; \boldsymbol{0}, \boldsymbol{I}). \tag{21}$$

where $\Delta t$ is a discrete time step. We aim to design this formula such that it becomes a Langevin discrete step that leads $\boldsymbol{a}^k$ to converge to $\mathcal{N}(\boldsymbol{0}, \boldsymbol{I})$ as $k \to \infty$. To achieve this, let

$$\boldsymbol{\mu}_\theta(\boldsymbol{a}^k, \boldsymbol{s}) = \frac{1}{2}\boldsymbol{\Sigma}_\theta(\boldsymbol{a}^k, \boldsymbol{s})\nabla_{\boldsymbol{a}^k}\log\mathcal{N}(\boldsymbol{a}^k; \boldsymbol{0}, \boldsymbol{I}) = \frac{1}{2}\boldsymbol{\Sigma}_\theta(\boldsymbol{a}^k, \boldsymbol{s})(-\boldsymbol{a}^k) \tag{22}$$

Substitue equation 22 into equation 21 and consider $\Delta t$ as a selected constant, we can regarded equation 21 is the discretization of the following SDE:

$$d\boldsymbol{a} = \frac{\boldsymbol{\Sigma}_\theta(\boldsymbol{a}, \boldsymbol{s})}{2\Delta t}(-\boldsymbol{a}) + \left(\frac{\boldsymbol{\Sigma}_\theta(\boldsymbol{a}, \boldsymbol{s})}{\Delta t}\right)^{\frac{1}{2}} d\boldsymbol{w} \tag{23}$$

In the practical implementation, $\boldsymbol{\Sigma}_\theta(\boldsymbol{a}^k, \boldsymbol{s})$ is always set being *independent of* $\boldsymbol{a}$ and $\boldsymbol{s}$, such as RSL-RL (Rudin et al., 2022). We adopt this common parameterization method in our method. In this case, we rewrite the SDE in equation 23 as

$$d\boldsymbol{a} = \frac{1}{2}\boldsymbol{G}\boldsymbol{G}^T\nabla_{\boldsymbol{a}}\log\mathcal{N}(\boldsymbol{a}; \boldsymbol{0}, \boldsymbol{I}) + \boldsymbol{G}d\boldsymbol{w} \qquad \boldsymbol{G} := \left(\frac{\boldsymbol{\Sigma}_\theta}{\Delta t}\right)^{\frac{1}{2}}, \tag{24}$$

which can be regarded as the Langevin dynamics with non-unit drift coefficients that drive the distribution into a standard Gaussian. Formally, this can be verified by the Fokker-Planck equation that the standard Gaussian is the stationary distribution:

$$0 = -\nabla \cdot (\boldsymbol{G}\boldsymbol{G}^T\nabla_{\boldsymbol{a}}\log\mathcal{N}(\boldsymbol{a}; \boldsymbol{0}, \boldsymbol{I})p^*(\boldsymbol{a})) + \nabla \cdot (\boldsymbol{G}\boldsymbol{G}^T\nabla_{\boldsymbol{a}}p^*(\boldsymbol{a})) \tag{25}$$

where $p^*(\boldsymbol{a}) = \mathcal{N}(\boldsymbol{a}; \boldsymbol{0}, \boldsymbol{I})$. Note $\boldsymbol{G}$ being independent of $\boldsymbol{a}$ is used. One potential problem is that regarding equation 21 being a discrete Euler-Maruyama step may introduce large numerical error due to $\boldsymbol{G}$ may be relatively large. However, in our implementation, we find that this will not cause a large error.

As a summary, by considering the policy iteration process as the diffusion generative process, this regularization term tends to let the whole policy iteration process be a Langevin dynamics that makes policy converge to a standard Gaussian $\mathcal{N}(\boldsymbol{0}, \boldsymbol{I})$.

# D EXPERIMENT DETAILS

## D.1 ISSACLAB

**Training Setup.** We evaluate the algorithms on IssacLab (IsaacSim 5.0.0). Its default installation includes the RSL-RL package, which is the PPO baseline used in this work. All dependencies are installed according to IsaacLab's documents. The training platform is one Nvidia-RTX-4090.

**Hyperparameters.** We adopt the default hyperparameters of RSL-RL PPO, which already provide strong performance across tasks. Our method consists of two components: (i) a conditional PPO (CPPO) for policy improvement, and (ii) a flow matching to fit the new diffusion policy. Since CPPO largely follows standard PPO, we keep the settings identical wherever possible and introduce only minor adjustments. The necessary adjustments are made because CPPO serves a different role than standard PPO, requiring different scaling. For the flow-matching component, we keep all parameters the same as in prior settings, except for the network size (see Tab. 5). The number of training epochs per policy iteration is deliberately kept small, as we find that increasing it does not improve performance much. We therefore set it to a suitable value to reduce computational cost. Other detailed parameters for each task are shown in Tab. 6 to 13.

**Results.** We evaluate the algorithms in eight tasks: Isaac-Ant-v0; Isaac-Lift-Cube-Franka-v0; Isaac-Quadcopter-Direct-v0; Isaac-Velocity-Flat-Anymal-D-v0; Isaac-Velocity-Rough-H1-v0; Isaac-Velocity-Rough-Unitree-Go2-v0; Isaac-Navigation-Flat-Anymal-C-v0; Isaac-Velocity-Rough-G1-v0. From the training curve Fig. 4, we find that the diffusion policy increases at a slightly slower rate than Gaussian PPO, which can be attributed to the diffusion model lying in an infinite functional space that is more difficult to train. For computational comparison in Tab. 2, we report PPO results from our own implementation instead of RSL-RL PPO, in order to maintain consistent experimental settings. This ensures that the percentage comparison with our method is more accurate and reliable.

## D.2 PLAYGROUND

**Training Setup.** We evaluate the algorithms on Playground 0.0.5. All dependencies are installed according to the Playground's default documents. The training platform is the same as in IsaacLab.

**Hyperparameters.** We adopt parameter settings similar to those in IsaacLab, with minor adjustments when necessary. The metric recording in FPO differs slightly from our convention; we have made them consistent for a fair comparison. The detailed parameters used in Playground are reported in Tab. 14. The network size we set is the same or less than FPO.

**Results.** We evaluate the algorithms in eight tasks: BallinCup; CheetahRun; FingerSpin; Point-Mass; ReacherEasy; ReacherHard; WalkerRun; WalkerWalk. In the comparison with FPO, since the data collection procedure per iteration differs significantly between the two methods, we report the reward curves with respect to the number of environment steps (see Fig. 5). Note that we extend the number of samples in CheetahRun and FingerSpin for our method, since in these two tasks the policy converges more slowly and requires additional training steps for the diffusion policy to stabilize. For fairness, however, we still report the results in Tab. 4 under their default settings.

## D.3 ADDITIONAL RESULTS

**Diffusion Policy Fitting Error.** In the proposed method, we utilize a single flow policy $\pi_{\text{flow}}$ to fit the optimal policy $\pi_{\theta*}(\boldsymbol{a}|\boldsymbol{s}) = \int \tilde{\pi}(\boldsymbol{a}_0|\boldsymbol{s})p_{\theta*}(\boldsymbol{a}|\boldsymbol{a}_0,\boldsymbol{s})d\boldsymbol{a}_0$ in each policy improvement. In Sec. 3.2, we have pointed out that this fitting will not introduce an accumulated error since every policy improvement is based on $\pi_{\text{flow}}$ rather than the $\pi_{\theta*}(\boldsymbol{a}|\boldsymbol{s})$. Here, we investigate how large this fitting error is in experiments and whether this error will impact the performance of the method. We present training curves for different numbers of flow-matching training epochs on the Isaac-Lift-Cube-Franka-v0 task, which exhibit only small variance across multiple runs. As shown in Fig. 7a, using fewer flow-matching epochs generally leads to larger fitting errors, reflected by lower rewards. However, these errors diminish as training progresses and the policy converges. From Fig. 7b, we observe that the primary effect of reduced flow-matching epochs is a slower improvement rate; the final performance converges to nearly the same reward across all settings. This indicates that our method is robust to the fitting errors induced by limited flow-matching training.

**Policy Monotonically Increasing.** In the proposed method, the policy monotonically increasing is approximately ensured by an EMA trick. In this section, we investigate whether monotonically increasing empirically holds and what the impact of EMA is. In the policy improvement phase, the policy is parameterized by $\pi_\theta(\boldsymbol{a}|\boldsymbol{s}) = \int \tilde{\pi}(\boldsymbol{a}_0|\boldsymbol{s})p_\theta(\boldsymbol{a}|\boldsymbol{a}_0,\boldsymbol{s})d\boldsymbol{a}_0$. The CPPO formulation can only improve the $\pi_{\theta_{\text{sampling}}}(\boldsymbol{a}|\boldsymbol{s})$ to $\pi_{\theta*}(\boldsymbol{a}|\boldsymbol{s})$, the relationship between $\pi_{\theta_{\text{sampling}}}(\boldsymbol{a}|\boldsymbol{s})$ and last diffusion policy $\tilde{\pi}(\boldsymbol{a}_0|\boldsymbol{s})$ is unclear. Thus, it is not theoretically guaranteed that $\pi_{\theta*}(\boldsymbol{a}|\boldsymbol{s})$ is better than $\tilde{\pi}(\boldsymbol{a}_0|\boldsymbol{s})$. To study this, we presents the reward difference of $\pi_{\theta_{\text{sampling}}}(\boldsymbol{a}|\boldsymbol{s})$ and $\tilde{\pi}(\boldsymbol{a}_0|\boldsymbol{s})$ in Fig. 8a. We observe that $\pi_{\theta_{\text{sampling}}}(\boldsymbol{a}|\boldsymbol{s})$ is consistently better than the reference policy $\tilde{\pi}(\boldsymbol{a}_0|\boldsymbol{s})$, providing empirical evidence that the proposed method achieves monotonic improvement under fitting errors and with EMA enabled. Note that the approximation in Eq. equation 13 describes the idealized case, whereas the experiment reflects the practical setting. In addition, we find that EMA greatly stabilizes training. Without EMA, the diffusion fitting stage often diverges, only 1 out of 6 runs succeeds. However, in the one successful trial, the final return is nearly the same as the case using EMA, as shown in Fig. 8b.

**Entropy and Score-Based Regularization.** We propose an entropy regularization to benefit exploration and an empirical score-based regularization term that improves optimization by preventing the policy from drifting too far from a prior distribution. Here, we provide a further joint ablation

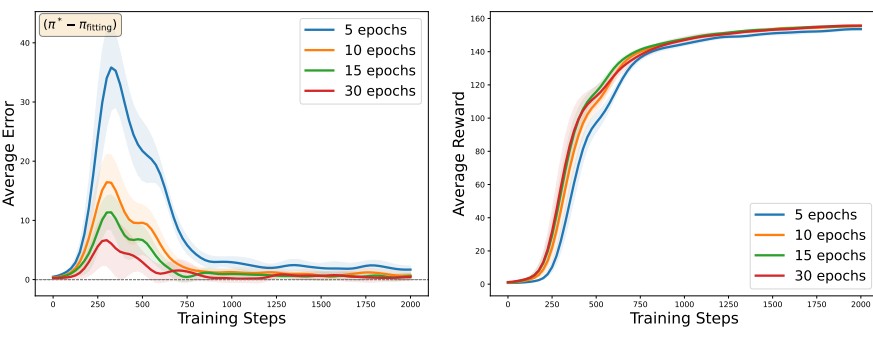

(a) Reward difference on Isaac-Lift-Cube-Franka-v0.     (b) Rewards on Isaac-Lift-Cube-Franka-v0.

Figure 7: Ablation study on the number of flow-matching training epochs. (a) Reward difference between the optimal policy and the flow-matched policy (denoted by $(\pi^* - \pi_{\text{fitting}})$), showing that the fitting error gradually decreases and converges to a small value. (b) Training rewards of the flow-matched policy across different epoch settings, all of which converge to nearly the same final performance.

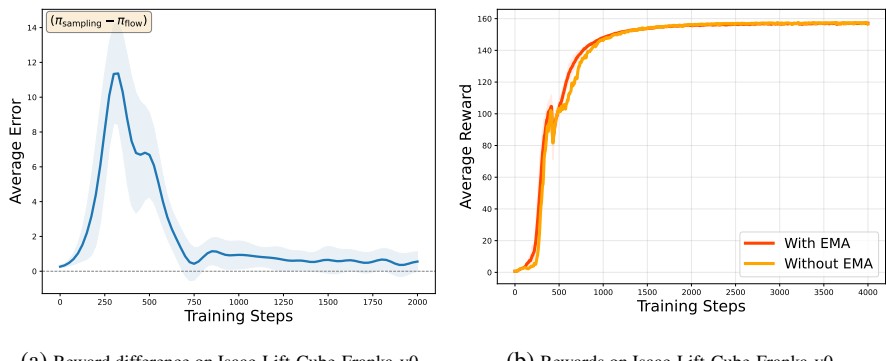

(a) Reward difference on Isaac-Lift-Cube-Franka-v0.     (b) Rewards on Isaac-Lift-Cube-Franka-v0.

Figure 8: (a) Reward difference between the sampling policy and the reference policy. The consistently positive values indicate that the sampling policy is better than the reference policy throughout training. (b) Training rewards with and without the EMA trick. Although the final performance is similar, the version without EMA succeeds in only 1 out of 6 runs, indicating it has reduced stability.

for score-based and entropy regularization to investigate how they impact each other. The results on Isaac-Quadcopter-Direct-v0 and Isaac-Velocity-Flat-Anymal-B-v0 tasks are shown in Fig. 9a and Fig. 9b, respectively. The training remains stable and converges more rapidly when both entropy regularization and score-based regularization are applied, indicating that these two regularizations complement each other. In the Quadcopter tasks, removing the score-based regularization leads to training failure due to policy divergence. In contrast, using only the score-based regularization (without entropy regularization) is sufficient for successful training, although the training process exhibits some oscillation. This highlights the strong stabilizing effect of the score-based term. In the Anymal-B task, training remains relatively stable across settings. However, without a score-based regularization, the reward improvement is significantly slower, and the policy becomes trapped in a local minimum. Interestingly, the score-based regularization alone enables the policy to escape this local minimum. A possible explanation is that locomotion in this environment admits a largely unimodal optimal policy, and encouraging the diffusion policy to remain close to a Gaussian prior improves convergence. Overall, we find that combining entropy regularization with score-based regularization consistently benefits the training process, though the full underlying mechanism needs further investigation.

Because entropy regularization maximizes only a lower bound of the true policy entropy, it is natural to ask how strongly maximizing this lower bound influences the actual entropy. In Fig. 10, we plot both the true entropy of the policy and its lower bound at the saddle point (0,0) over training epochs.

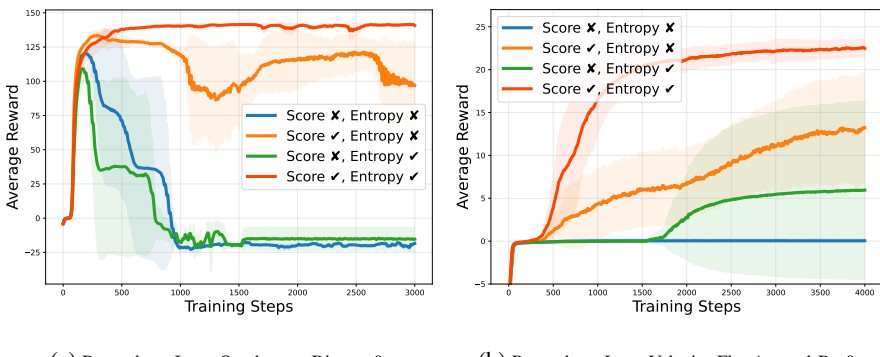

(a) Rewards on Isaac-Quadcopter-Direct-v0.

(b) Rewards on Isaac-Velocity-Flat-Anymal-B-v0.

Figure 9: Joint ablation on entropy and score-based regularization. In the figure, the check mark indicates that a suitable parameter is selected, while the cross indicates that this option is not applied.

The true entropy of the diffusion policy is numerically approximated using the mixed Gaussian method proposed by Wang et al. (2024). We observe that the two entropy curves exhibit generally consistent trends: increasing the weight of the lower-bound regularization reliably leads to higher true entropy. This holds both with and without score-based regularization. These results suggest that, in practice, maximizing the entropy lower bound is an effective surrogate for increasing the true policy entropy.

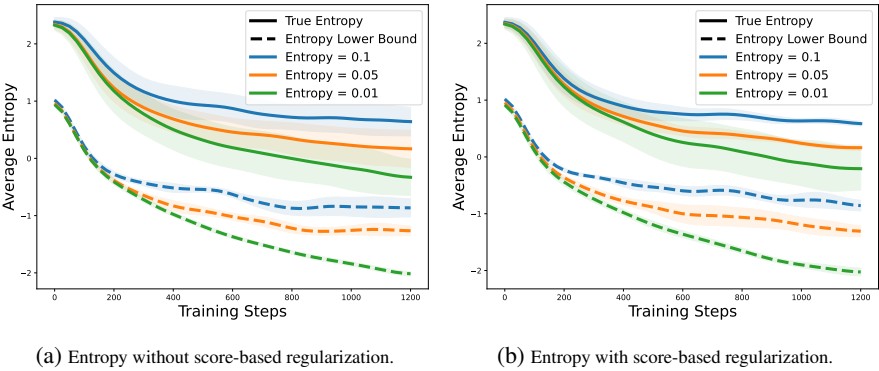

(a) Entropy without score-based regularization.

(b) Entropy with score-based regularization.

Figure 10: Entropy in Multi-Goal environment. The trends of the two entropy measures are generally consistent.

**Ablation on Flow Steps.** We also provide an ablation study on the number of flow-matching steps (i.e., ODE discretization steps). Since our method avoids embedding the recursive structure of the flow into the optimization gradient chain, the number of flow steps affects only the sampling quality and sampling speed during generation. We report the results on the Isaac-Lift-Cube-Franka-v0 task. As shown in Fig. 11, different flow steps result in nearly identical performance.

Table 5: Hyperparameters for Flow Matching (IsaacLab and Playground).

| Hyperparameters | Value |
| --- | --- |
| Flow steps | 16 |
| Learning rate | 0.001 |
| Ema rate | 0.999 |
| Mini batches | 4 |
| Training epochs | 15 |
| Optimizer | Adam |

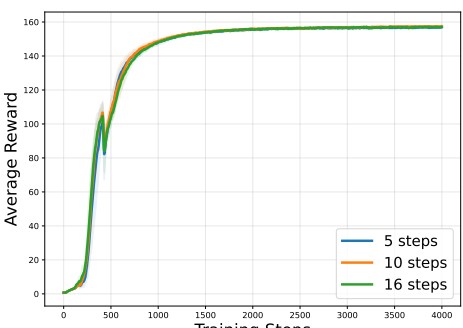

Figure 11: Reward on Isaac-Lift-Cube-Franka-v0 with different numbers of flow steps. All configurations achieve similar final performance.

Table 6: Hyperparameters for Isaac-Ant-v0.

| Hyperparameters | DP-CPPO | PPO |
|---|---|---|
| GAE Smoothing | 0.95 | 0.95 |
| Desired kl | 0.01 | 0.01 |
| Ratio clip | 0.2 | 0.2 |
| Max gradient clip | 1.0 | 1.0 |
| Value loss scale | 1.0 | 1.0 |
| Value clip | True | True |
| Num learning epochs | 5 | 5 |
| Num mini batches | 4 | 4 |
| LR schedule | adaptive | adaptive |
| Learning rate | 0.0005 | 0.0005 |
| Entropy loss scale | 0.005 | 0.01 |
| Score regularization loss scale | 0.1 | - |
| Rollouts | 32 | 32 |
| Num envs | 4096 | 4096 |
| Flow hidden dims | [400, 200, 100] | - |
| Gaussian policy hidden dims | [400, 200, 100] | [400, 200, 100] |
| Critic hidden dims | [400, 200, 100] | [400, 200, 100] |
| Activation | elu | elu |

Table 7: Hyperparameters for Isaac-Lift-Cube-Franka-v0.

| Hyperparameters | DP-CPPO | PPO |
|---|---|---|
| GAE Smoothing | 0.95 | 0.95 |
| Desired kl | 0.008 | 0.01 |
| Ratio clip | 0.2 | 0.2 |
| Max gradient clip | 1.0 | 1.0 |
| Value loss scale | 1.0 | 1.0 |
| Value clip | True | True |
| Num learning epochs | 5 | 5 |
| Num mini batches | 4 | 4 |
| LR schedule | adaptive | adaptive |
| Learning rate | 0.0001 | 0.0001 |
| Entropy loss scale | 0.004 | 0.006 |
| Score regularization loss scale | 0.1 | - |
| Rollouts | 24 | 24 |
| Num envs | 4096 | 4096 |
| Flow hidden dims | [256, 128, 64] | - |
| Gaussian policy hidden dims | [256, 128, 64] | [256, 128, 64] |
| Critic hidden dims | [256, 128, 64] | [256, 128, 64] |
| Activation | elu | elu |

Table 8: Hyperparameters for Isaac-Quadcopter-Direct-v0.

| Hyperparameters | DP-CPPO | PPO |
|---|---|---|
| GAE Smoothing | 0.95 | 0.95 |
| Desired kl | 0.01 | 0.01 |
| Ratio clip | 0.2 | 0.2 |
| Max gradient clip | 1.0 | 1.0 |
| Value loss scale | 1.0 | 1.0 |
| Value clip | True | True |
| Num learning epochs | 5 | 5 |
| Num mini batches | 4 | 4 |
| LR schedule | adaptive | adaptive |
| Learning rate | 0.0005 | 0.0005 |
| Entropy loss scale | 0.01 | 0.01 |
| Score regularization loss scale | 0.5 | - |
| Rollouts | 24 | 24 |
| Num envs | 4096 | 4096 |
| Flow hidden dims | [64, 64] | - |
| Gaussian policy hidden dims | [64, 64] | [64, 64] |
| Critic hidden dims | [64, 64] | [64, 64] |
| Activation | mish | elu |

Table 9: Hyperparameters for Isaac-Velocity-Flat-Anymal-D-v0.

| Hyperparameters | DP-CPPO | PPO |
|---|---|---|
| GAE Smoothing | 0.95 | 0.95 |
| Desired kl | 0.01 | 0.01 |
| Ratio clip | 0.2 | 0.2 |
| Max gradient clip | 1.0 | 1.0 |
| Value loss scale | 1.0 | 1.0 |
| Value clip | True | True |
| Num learning epochs | 5 | 5 |
| Num mini batches | 4 | 4 |
| LR schedule | adaptive | adaptive |
| Learning rate | 0.001 | 0.001 |
| Entropy loss scale | 0.001 | 0.005 |
| Score regularization loss scale | 0.1 | - |
| Rollouts | 24 | 24 |
| Num envs | 4096 | 4096 |
| Flow hidden dims | [128, 128, 128] | - |
| Gaussian policy hidden dims | [128, 128, 128] | [128, 128, 128] |
| Critic hidden dims | [128, 128, 128] | [128, 128, 128] |
| Activation | elu | elu |

Table 10: Hyperparameters for Isaac-Velocity-Rough-H1-v0.

| Hyperparameters | DP-CPPO | PPO |
|---|---|---|
| GAE Smoothing | 0.95 | 0.95 |
| Desired kl | 0.01 | 0.01 |
| Ratio clip | 0.2 | 0.2 |
| Max gradient clip | 1.0 | 1.0 |
| Value loss scale | 1.0 | 1.0 |
| Value clip | True | True |
| Num learning epochs | 5 | 5 |
| Num mini batches | 4 | 4 |
| LR schedule | adaptive | adaptive |
| Learning rate | 0.0005 | 0.001 |
| Entropy loss scale | 0.005 | 0.01 |
| Score regularization loss scale | 0.1 | - |
| Rollouts | 24 | 24 |
| Num envs | 4096 | 4096 |
| Flow hidden dims | [512, 256, 128] | - |
| Gaussian policy hidden dims | [512, 256, 128] | [512, 256, 128] |
| Critic hidden dims | [512, 256, 128] | [512, 256, 128] |
| Activation | elu | elu |

Table 11: Hyperparameters for Isaac-Velocity-Rough-Unitree-Go2-v0.

| Hyperparameters | DP-CPPO | PPO |
|---|---|---|
| GAE Smoothing | 0.95 | 0.95 |
| Desired kl | 0.01 | 0.01 |
| Ratio clip | 0.2 | 0.2 |
| Max gradient clip | 1.0 | 1.0 |
| Value loss scale | 1.0 | 1.0 |
| Value clip | True | True |
| Num learning epochs | 5 | 5 |
| Num mini batches | 4 | 4 |
| LR schedule | adaptive | adaptive |
| Learning rate | 0.001 | 0.001 |
| Entropy loss scale | 0.005 | 0.01 |
| Score regularization loss scale | 0.1 | - |
| Rollouts | 24 | 24 |
| Num envs | 4096 | 4096 |
| Flow hidden dims | [512, 256, 128] | - |
| Gaussian policy hidden dims | [512, 256, 128] | [512, 256, 128] |
| Critic hidden dims | [512, 256, 128] | [512, 256, 128] |

Table 12: Hyperparameters for Isaac-Navigation-Flat-Anymal-C-v0.

| Hyperparameters | DP-CPPO | PPO |
|---|---|---|
| GAE Smoothing | 0.95 | 0.95 |
| Desired kl | 0.01 | 0.01 |
| Ratio clip | 0.2 | 0.2 |
| Max gradient clip | 1.0 | 1.0 |
| Value loss scale | 1.0 | 1.0 |
| Value clip | True | True |
| Num learning epochs | 5 | 5 |
| Num mini batches | 4 | 4 |
| LR schedule | adaptive | adaptive |
| Learning rate | 0.001 | 0.001 |
| Entropy loss scale | 0.001 | 0.005 |
| Score regularization loss scale | 0.1 | - |
| Rollouts | 8 | 8 |
| Num envs | 4096 | 4096 |
| Flow hidden dims | [128, 128] | - |
| Gaussian policy hidden dims | [128, 128] | [128, 128] |
| Critic hidden dims | [128, 128] | [128, 128] |
| Activation | mish | elu |

Table 13: Hyperparameters for Isaac-Velocity-Rough-G1-v0.

| Hyperparameters | DP-CPPO | PPO |
|---|---|---|
| GAE Smoothing | 0.95 | 0.95 |
| Desired kl | 0.01 | 0.01 |
| Ratio clip | 0.2 | 0.2 |
| Max gradient clip | 1.0 | 1.0 |
| Value loss scale | 1.0 | 1.0 |
| Value clip | True | True |
| Num learning epochs | 5 | 5 |
| Num mini batches | 4 | 4 |
| LR schedule | adaptive | adaptive |
| Learning rate | 0.001 | 0.001 |
| Entropy loss scale | 0.001 | 0.008 |
| Score regularization loss scale | 0.1 | - |
| Rollouts | 24 | 24 |
| Num envs | 4096 | 4096 |
| Flow hidden dims | [512, 256, 128] | - |
| Gaussian policy hidden dims | [512, 256, 128] | [512, 256, 128] |
| Critic hidden dims | [512, 256, 128] | [512, 256, 128] |
| Activation | elu | elu |

Table 14: Hyperparameters for Playground tasks.

| Hyperparameters | Value |
| --- | --- |
| GAE Smoothing | 0.95 |
| Desired kl | 0.01 |
| Ratio clip | 0.2 |
| | 0.05 (BallinCup) |
| Max gradient clip | 1.0 |
| Value loss scale | 1.0 |
| Value clip | True |
| | False (BallinCup CheetahRun) |
| Num learning epochs | 5 |
| | 10 (CheetahRun FingerSpin) |
| Num mini batches | 4 |
| LR schedule | adaptive |
| | fix (PointMass) |
| Learning rate | 0.001 |
| | 0.0001 (BallinCup) |
| | 0.0005 (PointMass) |
| Entropy loss scale | 0.001 |
| | 0.005 (FingerSpin ReachEasy ReachHard) |
| Score regularization loss scale | 0.1 |
| Rollouts | 32 |
| Num envs | 4096 |
| Flow hidden dims | [128, 64, 32] |
| Gaussian policy hidden dims | [512, 256, 128] |
| Critic hidden dims | [512, 256, 128] |
| Activation | elu |

