# OpenReview forum: "Diffusion Policy through Conditional Proximal Policy Optimization"
_ICLR.cc/2026/Conference — Submitted to ICLR 2026_

### Official Review · Reviewer_xrap · 2025-10-27

**Soundness:** 2
**Presentation:** 1
**Contribution:** 2
**Rating:** 2
**Confidence:** 4

**Summary:**

This paper proposes Diffusion Policy through Conditional Proximal Policy Optimization (DP-CPPO), a novel on-policy reinforcement learning (RL) framework that integrates diffusion generative models with Proximal Policy Optimization (PPO). The key motivation stems from the difficulty of computing action log-likelihoods within diffusion models, which traditionally limits their use in on-policy settings.

**Strengths:**

The method avoids the costly recursive log-likelihood computation typical in diffusion-based RL (GenPo or DACER), replacing it with a simple Gaussian-based PPO step.

Unlike prior diffusion methods (FPO), which struggle with entropy terms, DP-CPPO supports entropy regularization analytically through a Gaussian lower bound, enabling controlled exploration in policy learning.

**Weaknesses:**

- The explanation at the end of 3.2 is far-fetched. EMA is a parameter smoothing but not a KL constraint.
* Missing comparisons with recent **on-policy diffusion methods** such as GenPo and DPPO.
* Absent results on **off-policy diffusion-RL methods** (**DIME**, **DACER**, **MaxEntDP**) evaluated on OpenAI Gym MuJoCo benchmarks.
* Multimodal behaviors are only demonstrated qualitatively in a **Multi-Goal** environment; **no quantitative metrics** (e.g., KL-divergence, mode count, action entropy) are reported.
* No **wall-clock performance comparison** with **FPO** under equivalent computational resources, making the claim of “high computational efficiency” unsubstantiated.
* No **variance statistics across random seeds** (only mean and visualization curves are reported), making it difficult to assess robustness under high-variance RL training.

Ablations

**Regularization Terms:** Independently and jointly ablate the entropy and score regularizations.

**EMA Mechanism:** Evaluate performance degradation when EMA is removed.

**Flow Steps:** Conduct a systematic study on how the number of flow steps affects the reward.

**Questions:**

How tight is the bound of this entropy related to mutual information? How will this error affect the experimental parameter setting? Can you provide an experimental analysis?

Given that the monotonic improvement of the strategy cannot be strictly guaranteed at the end of Section 3.2, is it reasonable to regard the diffusion process as a policy iteration process?

---

> ### Author Response · Authors · 2025-11-22
>
> We thank the reviewer for their careful review and thoughtful comments. We have addressed the main concerns in the revised manuscript, which has now been uploaded to the system. Please see our detailed responses below.
>
> > The explanation at the end of 3.2 is far-fetched. EMA is a parameter smoothing but not a KL constraint.
>
> We thank the reviewer for pointing this out. The limited magnitude of policy updates is a consequence of using EMA, rather than EMA imposing an actual KL constraint on the optimization problem. We have removed the previous explanation in the revised manuscript.
>
> > Missing comparisons with recent on-policy diffusion methods such as GenPo and DPPO.
>
> Thanks for the suggestion. GenPo is not open-sourced currently; it is challenging to compare with it. We are currently implementing DPPO to include it in the comparison.
>
> > Absent results on off-policy diffusion-RL methods (DIME, DACER, MaxEntDP) evaluated on OpenAI Gym MuJoCo benchmarks.
>
> Our main focus in this work is on on-policy algorithms, and it is non-trivial to conduct a fair comparison against off-policy methods. More importantly, the baselines we include already capture the essential comparisons needed to validate our core claims. In addition, the OpenAI Gym and MuJoCo benchmarks do not support scalable parallel execution, which is a key requirement for on-policy algorithms to work effectively.
>
> > Multimodal behaviors are only demonstrated qualitatively in a Multi-Goal environment; no quantitative metrics (e.g., KL-divergence, mode count, action entropy) are reported.
>
> Thanks for the suggestion. We have added a mode count in the revised manuscript (Sec. 4.1) to quantitatively demonstrate the multi-modality of the proposed diffusion policy.
>
> > No wall-clock performance comparison with FPO under equivalent computational resources, making the claim of “high computational efficiency” unsubstantiated.
>
> Our computational efficiency claim is based on the method that embeds the diffusion recursive structure into the policy optimization, like GenPo. In contrast, our main point regarding FPO is that it can not deal with entropy regularization. Therefore, we do not provide a computational comparison with FPO.
>
> > No variance statistics across random seeds (only mean and visualization curves are reported), making it difficult to assess robustness under high-variance RL training.
>
> Thanks for the suggestion. We have added a standard deviation result to the table to assess the robustness of the method. Please see Tab. 3 and 4 in the revised manuscript.
>
> > Ablations. Regularization Terms: Independently and jointly ablate the entropy and score regularizations.
> EMA Mechanism: Evaluate performance degradation when EMA is removed.
> Flow Steps: Conduct a systematic study on how the number of flow steps affects the reward.
>
> Thanks for the suggestion. We can include these ablations in the revised manuscripts. Please find them in Appendix D3.
>
>
> > How tight is the bound of this entropy related to mutual information? How will this error affect the experimental parameter setting? Can you provide an experimental analysis?
>
> Thanks for the question. We have added an additional experiment addressing this point in the revised manuscript (Appendix D3). The results show that the trend of the lower bound closely matches that of the true policy entropy. Moreover, applying different weights to the lower bound leads to effective adjustment of the original entropy during training.
>
>
> > Given that the monotonic improvement of the strategy cannot be strictly guaranteed at the end of Section 3.2, is it reasonable to regard the diffusion process as a policy iteration process?
>
> Thanks for the question. It can not rigorously claim that they are equivalent in the proposed method; we only think about how to build a framework to align them together. The lack of a monotonic improvement guarantee for the policy is a limitation of this method yet. However, in the experiments, we observe that the monotonic improvement property empirically holds. Please see Appendix D3 for details.

---

> > ### Author Response · Authors · 2025-12-01
> >
> > Now, we have included the results of DPPO in the revised manuscript; our method still gives a superior result, please see Sec. 4.3 for details.

---

### Official Review · Reviewer_ynb6 · 2025-10-28

**Soundness:** 3
**Presentation:** 3
**Contribution:** 3
**Rating:** 6
**Confidence:** 4

**Summary:**

This paper presents a novel and efficient framework to train diffusion policies in on-policy reinforcement learning. Instead of directly optimizing diffusion policies, this paper aligns the policy iteration process with the diffusion process. Specifically, at each policy improvement step,  the old diffusion policy is forzen and a Gaussian residual policy is trained via conditional PPO to optimize the combination of these two models. Then, the combination policy is distilled into a new diffusion policy using the flow matching loss. The old diffusion policy is updated utilizing the EMA technique, and the proposed policy iteration can be approximately regarded as monotonically improving. Additionally, entropy regularization and score-based regularization are incorporated to enhance exploration and training stability, respectively. Experiments on IssacLab and MuJoCo Playgroud demonstrate that the proposed method effectively learns multimodal policies and achieve superior performance across various benchmark tasks.

**Strengths:**

1. The paper addresses a key and challenging problem—applying diffusion policies in on-policy reinforcement learning.
2. The proposed training framework is both efficient and elegant, as it only requires optimizing the Gaussian residual policy and training diffusion models using the simple flow matching loss, thereby avoiding the intractable computation of diffusion model log-likelihoods.

**Weaknesses:**

1. The paper would be strengthened by including a comparison with DPPO[1], which also employs an on-policy RL algorithm (PPO) to train diffusion policies.
2. Moving the learning curves from the appendix to the main text would provide a clearer comparison of performance against baseline methods.

**Questions:**

1. In line 268, the paper states that the score-based regularization term, which tends to let a Langevin dynamics update toward the standard Gaussian, can accelerate and stabilize training. Why such a term can accelerate and stabilize training?
2. Is the proposed training paradigm—aligning policy iteration with the diffusion process—also applicable to off-policy reinforcement learning? Have the authors experimented with replacing PPO with off-policy algorithms when training the Gaussian residual policy?

[1] Allen Z Ren, Justin Lidard, Lars L Ankile, Anthony Simeonov, Pulkit Agrawal, Anirudha Majumdar, Benjamin Burchfiel, Hongkai Dai, and Max Simchowitz. Diffusion Policy Policy Optimization. International Conference on Learning Representation. 2025.

---

> ### Author Response · Authors · 2025-11-22
>
> We thank the reviewer for their careful review and thoughtful comments. We have addressed the main concerns in the revised manuscript, which has now been uploaded to the system. Please see our detailed responses below.
>
> > The paper would be strengthened by including a comparison with DPPO[1], which also employs an on-policy RL algorithm (PPO) to train diffusion policies.
>
> Thanks for the suggestion. We are currently working on implementing DPPO to include it in our comparison.
>
> > Moving the learning curves from the appendix to the main text would provide a clearer comparison of performance against baseline methods.
>
> Thanks for the suggestion. We have moved the learning curves in the main text in the revised manuscript.
>
> > In line 268, the paper states that the score-based regularization term, which tends to let a Langevin dynamics update toward the standard Gaussian, can accelerate and stabilize training. Why such a term can accelerate and stabilize training?
>
> Thanks for the question. This term is like a KL-like constraint to the diffusion policy, preventing the policy not drifting far away from a prior distribution. This is only an empirical choice; the true mechanism remains an open question now. But we can share our motivation (or possible explanation). The diffusion policy's searching space is much larger than a Gaussian policy; it can move into unusual regions, leading to instability or divergence during training. A prior distribution can somehow mitigate this, so it stabilizes the training process. For the accelerating effect, in some tasks, the optimal policy may be only a unimodal distribution. In such cases, a Gaussian prior can guide the policy toward this unimodal structure, thereby speeding up training. We also provide an additional ablation study on this term in the revised manuscript (Appendix D3).
>
> > Is the proposed training paradigm—aligning policy iteration with the diffusion process—also applicable to off-policy reinforcement learning? Have the authors experimented with replacing PPO with off-policy algorithms when training the Gaussian residual policy?
>
> Thanks for the suggestion. We believe the proposed method is compatible with an off-policy setting, keeps the same policy parameterization, and simply replaces the CPPO update with an appropriate off-policy optimization objective. In this work, our primary goal is to demonstrate the effectiveness of the method, so we focus on the more straightforward on-policy setting. But extending the framework to off-policy methods is a promising direction for future work.

---

> > ### Author Response · Authors · 2025-12-01
> >
> > Now, we have included the results of DPPO in the revised manuscript; our method still gives a superior result, please see Sec. 4.3 for details.

---

### Official Review · Reviewer_UeJD · 2025-10-31

**Soundness:** 3
**Presentation:** 4
**Contribution:** 4
**Rating:** 8
**Confidence:** 4

**Summary:**

This paper proposes a novel policy parameterization scheme and training algorithm. By parameterizing the new policy as a convolution of a reference policy and a conditional Gaussian kernel , the policy optimization process and the entropy regularization step are made simple and efficient. The authors demonstrate the algorithm's stability and effectiveness through extensive experiments.

Main Contributions:

A Novel DP-CPPO Framework: The authors propose a new reinforcement learning framework (DP-CPPO) that effectively supports the use of diffusion models in on-policy algorithms and, notably, is compatible with entropy regularization.

Innovative Policy Parameterization: By parameterizing the new policy as a convolution of a reference policy and a conditional Gaussian kernel, the policy optimization process and entropy regularization are made simple and efficient.

Tractable Entropy Regularization: The framework naturally resolves the difficulty of computing the diffusion policy's entropy $\mathcal{H}(\pi_{\theta})$. The authors achieve efficient exploration by maximizing a tractable lower bound of the entropy, namely the entropy of the Gaussian.

**Strengths:**

The main strength of this paper lies in its novel and significant methodology, which cleverly bypasses the intractable $log \pi(a|s)$ computation in on-policy diffusion training by treating each policy iteration as a denoising step. The method is computationally efficient, with GPU memory occupation comparable to PPO while maintaining reasonable training times. Furthermore, it elegantly solves the difficult entropy regularization problem by optimizing a tractable entropy lower bound, a key feature lacking in methods like FPO. Strong empirical results, including demonstrated multi-modal capabilities, excellent benchmark performance, and ablation studies proving the necessity of all components, confirm the method's effectiveness.

**Weaknesses:**

The method has several weaknesses. First, it introduces a policy fitting step (Flow Matching) after the optimization step (CPPO), which creates an approximation error whose cumulative impact on convergence is unassessed. Second, the algorithm relies heavily on an EMA approximation to ensure monotonic improvement, which is not a theoretical guarantee and may fail if policy updates are too large.

**Questions:**

1. Regarding stability, how does the Flow Matching fitting error behave during training? What impact does this error have on the stability of the CPPO optimization step? Is there a risk of the fitting process lagging behind large policy updates?

2. What is the effective batch size (or number of samples) used to update the flow model (Eq. 12) in each policy iteration?

3. Instead of training the flow model for a fixed number of epochs in each iteration (Algorithm 1, Line 4-5), have you considered an adaptive update scheme? For instance, training the flow model until its loss (Eq. 12) converges below a specific threshold. What effect might this have on the overall algorithm's stability and computational efficiency?

---

> ### Author Response · Authors · 2025-11-22
>
> We thank the reviewer for their careful review and thoughtful comments. We have addressed the main concerns in the revised manuscript, which has now been uploaded to the system. Please see our detailed responses below.
>
> > First, it introduces a policy fitting step (Flow Matching) after the optimization step (CPPO), which creates an approximation error whose cumulative impact on convergence is unassessed.
>
> We agree with the reviewer that this is an important point. The fitting step indeed introduces an error in fitting the new policy, we have added an ablation study in the revised manuscript to analyze its effect on performance (see Appendix D3). Besides, we would like to clarify two points that may help address this concern. First, this fitting error will not be cumulative into the next policy iterations, since the policy improvement aims to improve the fitting policy rather than the previous optimal policy, which ensures the error will not be cumulated in the whole process. Second, we observe empirically that the fitting error is relatively small, and our method is robust even when the error is relatively large, with final performance nearly unchanged.
>
> > Second, the algorithm relies heavily on an EMA approximation to ensure monotonic improvement, which is not a theoretical guarantee and may fail if policy updates are too large.
>
> This is a limitation of this method. The theoretical monotonic improvement is not guaranteed in the current version. However, in our experiments, we observe that the monotonic improvement generally empirically holds. To demonstrate this, we add an additional ablation study on this problem in the revised manuscript. Please see Appendix D3 for details.
>
> > Regarding stability, how does the Flow Matching fitting error behave during training? What impact does this error have on the stability of the CPPO optimization step? Is there a risk of the fitting process lagging behind large policy updates?
>
> This fitting error is small in our experiments, mainly due to every policy improvement step only updating the policy modestly (following the RSL-RL PPO setting). Moreover, in our experiments, we find that this error (or flow policy update-lagging) will converge soon and will not impact the final performance much. We think the proposed method is robust to this fitting error. Please refer to Appendix D3 in the revised manuscript for details.
>
> > What is the effective batch size (or number of samples) used to update the flow model (Eq. 12) in each policy iteration?
>
> The number of samples is the same as that used in the CPPO problem, that is, num-env (typically 4096) * rollout (typically 32) = 131,072 samples for each flow matching update in one policy iteration.
>
> > Instead of training the flow model for a fixed number of epochs in each iteration (Algorithm 1, Line 4-5), have you considered an adaptive update scheme? For instance, training the flow model until its loss (Eq. 12) converges below a specific threshold. What effect might this have on the overall algorithm's stability and computational efficiency?
>
> Thanks for the suggestion. This is a good idea to control the fitting error throughout the whole process. Currently, we find the proposed method is empirically robust to this fitting error (see our responses A1 and A3). Thus, for simplicity, we adopt a fixed number of training epochs in the current implementation. That said, designing an adaptive update scheme could further improve convergence and stability, and we consider this a promising direction for future work.

---

### Official Review · Reviewer_wG1b · 2025-11-01

**Soundness:** 3
**Presentation:** 3
**Contribution:** 3
**Rating:** 6
**Confidence:** 4

**Summary:**

This paper proposes a diffusion-based on-policy RL method. Concretely, given an existing policy \(\pi^{k}\), a single-step Gaussian kernel is trained using the PPO objective (Equation (11)) such that \(\pi^{k}\), when passed through the kernel, produces a target distribution \(\pi^{\mathrm{target}}\). Subsequently, Flow Matching is used to fit this target distribution and thereby complete training. In addition, the authors investigate score-based regularization of the Gaussian kernel to align it with a single diffusion step; ablation experiments reveal its efficacy. The work is evaluated on benchmarks such as MuJoCo Playground and IsaacLab, and demonstrates top performance compared to other baselines.

**Strengths:**

1. The method is simple yet novel: it separates the Flow Matching stage from reinforcement-learning policy improvement, by first training a Teacher policy under a familiar PPO training paradigm and then using Flow Matching to learn that Teacher policy.
2. The ablation studies are thorough: through experiments the paper convincingly shows the importance of both the entropy regularization (which allows the method to outperform FPO) and the score-based regularization (which prevents the Teacher policy from diverging and thus makes Flow Matching feasible).

**Weaknesses:**

1. Some experimental descriptions are insufficiently clear. The meaning of “Flow” vs. “Flow + Residual” is ambiguous. The original text only uses the phrase *“diffusion-only (denotes ‘Flow’) policy and the combined policy”* (lines 390–391) without further clarifying exactly what “Residual” constitutes.
2. Details of the training process are missing. Since the method relies on Flow Matching to fit \(\pi^{k}\) after \(p_{\boldsymbol{\theta}}\), the paper should report: evidence that Flow Matching converges to the Teacher policy. Although Table 5 reports `training epochs = 15` and `mini batches = 4`, I am concerned that this is insufficient to guarantee that the Flow-Matching stage has converged to the Teacher policy.
3. Baselines are too few. The paper uses only FPO and PPO as baselines, which limits the strength of the empirical claims. Even though the authors mention (line 382) the implementation difficulty of FPO in a Torch-based IsaacLab environment, the omission of other relevant baselines is still a drawback. For example, the following open-source diffusion-RL baselines are available:

   * DACER ([https://github.com/happy-yan/DACER-Diffusion-with-Online-RL](https://github.com/happy-yan/DACER-Diffusion-with-Online-RL)) – JAX implementation
   * QVPO ([https://github.com/wadx2019/qvpo](https://github.com/wadx2019/qvpo)) – PyTorch implementation
   * DPPO ([https://github.com/irom-princeton/dppo](https://github.com/irom-princeton/dppo)) – PyTorch implementation
   * DIPO ([https://github.com/BellmanTimeHut/DIPO](https://github.com/BellmanTimeHut/DIPO)) – PyTorch implementation
     Given the availability of these implementations, it seems feasible to include them as baselines.

**Questions:**

In Section 3.3 you present score-based regularization and show its empirical benefit. My question is: why is it necessary to align the one-step Gaussian kernel with a one-step diffusion update? More specifically, would a simpler regularizer such as \(\mathrm{KL}(p_\theta||p_{\theta_{old}})\) (to prevent the kernel from making large jumps) suffice? In other words, the score-based regularizer seems somewhat analogous to enforcing a TRPO-style trust region; is the more sophisticated “score alignment” strictly required?

---

> ### Author Response · Authors · 2025-11-22
>
> We thank the reviewer for their careful review and thoughtful comments. We have addressed the main concerns in the revised manuscript, which has now been uploaded to the system. Please see our detailed responses below.
>
> > Some experimental descriptions are insufficiently clear. The meaning of “Flow” vs. “Flow + Residual” is ambiguous. The original text only uses the phrase “diffusion-only (denotes ‘Flow’) policy and the combined policy” (lines 390–391) without further clarifying exactly what “Residual” constitutes.
>
> Thanks for the suggestion. The parameterized policy in policy improvement step is given by $\pi_\theta(a|s)=\int\tilde{\pi}(a_0|s)p_\theta(a|a_0,s)da_0$, the "Flow" policy means $\tilde{\pi}$, the "Flow + Residual" means $\pi_\theta$. We have clarified this in the revised manuscript, see Sec. 4.3.
>
> > Details of the training process are missing. Since the method relies on Flow Matching to fit $(\pi^{k})$ after $(p_{\boldsymbol{\theta}})$, the paper should report: evidence that Flow Matching converges to the Teacher policy. Although Table 5 reports training epochs = 15 and mini batches = 4, I am concerned that this is insufficient to guarantee that the Flow-Matching stage has converged to the Teacher policy.
>
> Thanks for the suggestion. The flow matching process for fitting the teacher policy does introduce some numerical error; however, we find this error will not impact the performance much from the following two points. First, every policy improvement step aims to improve the current flow policy rather than the last teacher policy, which prevents accumulating the error into the next policy iteration. Second, in the experiments, we found that this error is not large since the teacher policy changes only modestly at each update (following the RSL-RL PPO setting), making the fitting error quite small in practice (in the paper's setting). So the setting in the paper (training epochs=15 and minibatches=4) is enough for every improvement. Even when the error becomes relatively larger, such an error will converge soon and will not impact the final performance. We have added an ablation study to illustrate this effect. Please refer to Appendix D3 in the revised manuscript.
>
>
> > Baselines are too few. The paper uses only FPO and PPO as baselines, which limits the strength of the empirical claims. Even though the authors mention (line 382) the implementation difficulty of FPO in a Torch-based IsaacLab environment, the omission of other relevant baselines is still a drawback. For example, the following open-source diffusion-RL baselines are available:
> DACER (https://github.com/happy-yan/DACER-Diffusion-with-Online-RL) – JAX implementation
> QVPO (https://github.com/wadx2019/qvpo) – PyTorch implementation
> DPPO (https://github.com/irom-princeton/dppo) – PyTorch implementation
> DIPO (https://github.com/BellmanTimeHut/DIPO) – PyTorch implementation Given the availability of these implementations, it seems feasible to include them as baselines.
>
> Thanks for the suggestion. We agree that additional baselines could further strengthen our work.
> Many related baselines follow the off-policy setting, which differs from our on-policy setting. It is not trivial to conduct a fair comparison. We would like to state that our current baselines already capture the essential comparisons needed to validate the core claims. However, it would be more convincing to include more baselines. So we are currently working on implementing DPPO, which is the most related work to ours.
>
> > In Section 3.3 you present score-based regularization and show its empirical benefit. My question is: why is it necessary to align the one-step Gaussian kernel with a one-step diffusion update? More specifically, would a simpler regularizer such as $(\mathrm{KL}(p_\theta||p_{\theta_{old}}))$ (to prevent the kernel from making large jumps) suffice? In other words, the score-based regularizer seems somewhat analogous to enforcing a TRPO-style trust region; is the more sophisticated “score alignment” strictly required?
>
> Thanks for the question. Although this regularization term looks like it prevents a large jump on the Gaussian kernel, it is actually the clipping trick that directly enforces that constraint. Instead, this term functions more like a KL-style regularizer on the *diffusion policy*, preventing it from drifting too far from a prior distribution. We include additional ablation results for this component in the revised manuscript (see Appendix D3). The motivation for introducing this term is that we observed that the training is sometimes not stable and can not increase rapidly. We want to assign a prior regularization to the policy to stabilize and accelerate the training. The current score-based regularization is only one empirical option; we believe there may exist better regularization, which can be the future work.

---

> > ### Author Response · Authors · 2025-12-01
> >
> > Now, we have included the results of DPPO in the revised manuscript; our method still gives a superior result, please see Sec. 4.3 for details.

---

### Public Comment · ~Haque_Ishfaq1 · 2025-11-15
**Comparing against Langevin Soft Actor Critic  from ICLR 2025**

Dear authors,

This seems like a really nice work. We wanted to point out that our ICLR 2025 paper on Langevin Soft Actor Critic [1] should be used as a baseline. In LSAC, we use Langevin Monte Carlo (LMC) based Thompson Sampling and distributional critic to achieve sample efficiency as well as multi-modal behavior through parallel tempering based LMC. Given one of the main motivations for this submission is achieving multi-modal behavior in the policy, we believe our LSAC paper is highly relevant as an alternative to diffusion policy based approaches.

In our work, we used `DSAC-T` (Duan et al., 2023), `QSM` (Psenka et al., 2024), `DIPO` (Yang et al., 2023), `SAC` (Haarnoja et al., 2018a), `TD3` (Fujimoto et al., 2018), `PPO` (Schulman et al., 2017), `TRPO` (Schulman et al., 2015), `REDQ` (Chen et al., 2021),  `DrQ-v2` (Yarats et al., 2022)) and model-based (`Dreamer` (Hafner et al., 2020)) as baselines and we got superior results than these baselines in MuJoCo tasks and DeepMind Control Suite tasks. The aforementioned baselines also include some of the requested baselines mentioned by *Reviewer wG1b*.

Our codebase and all the data are publicly available here https://github.com/hmishfaq/LSAC

We would appreciate if you could provide a comparison against LSAC in your experiments.

Thanks!

[1] Ishfaq, Haque, et al. "Langevin Soft Actor-Critic: Efficient Exploration through Uncertainty-Driven Critic Learning."  ICLR 2025

---

> ### Author Response · Authors · 2025-11-22
>
> Thank you for your interest in our work and for sharing LSAC. We appreciate the pointer and agree that LSAC is highly relevant. At this stage, however, our experiments focus mainly on on-policy methods, and making a fair comparison with off-policy approaches is nontrivial, so incorporating such baselines is beyond the scope of our current submission. We will keep LSAC in mind for future extensions of our work.
>
> Thanks again for your comment.

---

### Meta-Review · Area_Chair_2oBY · 2025-12-25

**Summary:**

The paper proposes DP-CPPO, an on-policy framework for training diffusion policies. While the motivation to handle entropy regularization in diffusion RL is valid, the execution and validation fall short of conference standards. Reviewers identified critical deficiencies, including the lack of theoretical guarantees for the proposed policy iteration scheme and insufficient experimental comparisons. Crucially, the authors declined to provide requested comparisons against established off-policy diffusion RL baselines, limiting the assessment of the method's competitiveness. Due to these unresolved issues regarding soundness and comprehensive validation, the paper is recommended for rejection.

**Reviewer Concerns:**

While the authors provided some additional data (e.g., DPPO baseline, mode counts), key concerns raised by reviewers remain largely unaddressed or were met with insufficient justification.

*   **Addressed Concerns:**
    *   **DPPO Baseline:** The authors added a comparison with DPPO as requested by multiple reviewers.
    *   **Clarifications:** Some definitions were clarified, and the incorrect theoretical claim regarding EMA was removed.

*   **Outstanding Concerns:**
    *   **Insufficient Baselines:** Reviewer xrap requested comparisons with standard off-policy diffusion RL methods (e.g., DACER, DIME) to benchmark performance against the broader state-of-the-art. The authors declined these experiments, arguing they were out of scope. However, demonstrating superiority or at least competitiveness against the best available diffusion-based methods (regardless of on/off-policy classification) is essential to validate the proposed method's practical value.
    *   **Theoretical Soundness:** A fundamental issue raised by Reviewers UeJD and xrap is the lack of theoretical guarantees for monotonic policy improvement. The proposed method relies on an EMA heuristic to align the diffusion process with policy iteration. The authors acknowledged this lack of guarantee but did not provide a rigorous theoretical fix, leaving the algorithm's convergence properties in question.
    *   **Stability and Fitting Error:** The reliance on a Flow Matching step introduces an approximation error at every iteration. While the authors provided ablations, the concern regarding the long-term stability of this "fitting-then-optimizing" loop without theoretical backing remains valid.

**Reviewer Scores:**

*   **Reviewer wG1b:** 6 -> 6. The addition of the DPPO baseline is insufficient to change the initial "marginal" assessment.
*   **Reviewer UeJD:** 8 -> 8. The reviewer remains positive about the methodology and satisfied with the clarifications on stability.
*   **Reviewer xrap:** 2 -> 2. The rebuttal failed to provide the requested off-policy comparisons and did not resolve fundamental theoretical concerns.
*   **Reviewer ynb6:** 6 -> 6. The revisions are viewed as incremental and do not significantly enhance the paper's contribution to justify a higher score.

---

### Decision · Program_Chairs · 2026-01-26

Reject